# Monolayer platform to generate and purify primordial germ-like cells in vitro provides insights into human germline specification

Sivakamasundari Vijayakumar[1,2,5], Roberta Sala[1,3,5], Gugene Kang[1,3,5], Angela Chen[1,2], Michelle Ann Pablo[1,3], Abidemi Ismail Adebayo[1,3,4], Andrea Cipriano[1,3], Jonas L. Fowler[1,2], Danielle L. Gomes[1,3], Lay Teng Ang [1], Kyle M. Loh [1,3,6] ✉ & Vittorio Sebastiano [1,3,6] ✉

Generating primordial germ cell-like cells (PGCLCs) from human pluripotent stem cells (hPSCs) advances studies of human reproduction and development of infertility treatments, but often entails complex 3D aggregates. Here we develop a simplified, monolayer method to differentiate hPSCs into PGCs within 3.5 days. We use our simplified differentiation platform and single-cell RNA-sequencing to achieve further insights into PGCLC specification. Transient WNT activation for 12 h followed by WNT inhibition specified PGCLCs; by contrast, sustained WNT induced primitive streak. Thus, somatic cells (primitive streak) and PGCLCs are related—yet distinct—lineages segregated by temporally-dynamic signaling. Pluripotency factors including NANOG are continuously expressed during the transition from pluripotency to posterior epiblast to PGCs, thus bridging pluripotent and germline states. Finally, hPSC-derived PGCLCs can be easily purified by virtue of their CXCR4⁺PDGFRA⁻GARP⁻ surface-marker profile and single-cell RNA-sequencing reveals that they harbor transcriptional similarities with fetal PGCs.

Within the mammalian embryo, primordial germ cells (PGCs) are the harbinger to eggs and sperm; consequently, they are key to the act of reproduction and the vertical transmission of genetic and epigenetic information to the next generation. The developmental origins of mouse PGCs have been thoroughly explored[1–3], thus enabling the stepwise differentiation of mouse pluripotent cells into PGC-like cells (PGCLCs) in vitro in 3D cultures[4,5]. Pluripotent stem cells (PSCs) from both humans and non-human primates have likewise been successfully differentiated into PGCLCs in 3D cultures[6–16]. The ability to generate human PGCLCs in vitro has shed light on early human germline specification as well as genetic diseases such as infertility[17,18] and could eventually enable in vitro manufacture of human eggs and sperm for infertility treatments and other reproductive technologies.

Fundamentally speaking, the developmental precursors to human PGCs and the signals that specify their formation in vivo have remained hitherto uncertain because PGCs arise in weeks 2-3 of human embryogenesis[19,20] and it is ethically and technically difficult to attain and analyze early post-implantation human embryos. Consequently, knowledge is inferred from in vivo analyses of embryos from related species as well as in vitro differentiation of human PSCs (hPSCs) into PGCLCs[21]. Immediately prior to gastrulation, the pluripotent epiblast (which corresponds to PSCs) undergoes anterior-posterior patterning, generating anterior and posterior epiblast regions[8,22]. Subsequently, in

[1]Institute for Stem Cell Biology & Regenerative Medicine, Stanford University School of Medicine, Stanford, CA 94305, USA. [2]Department of Developmental Biology, Stanford University School of Medicine, Stanford, CA 94305, USA. [3]Department of Obstetrics & Gynecology, Stanford University School of Medicine, Stanford, CA 94305, USA. [4]Department of Mechanical Engineering, Stanford University, Stanford, CA 94305, USA. [5]These authors contributed equally: Sivakamasundari Vijayakumar, Roberta Sala, Gugene Kang. [8]These authors jointly supervised this work: Kyle M. Loh, Vittorio Sebastiano. ✉e-mail: kyleloh@stanford.edu; vsebast@stanford.edu

mouse and pig embryos, PGCs originate in the vicinity of the posterior epiblast; the posterior epiblast also gives rise to the primitive streak (the precursor to endoderm and mesoderm, Fig. 1a)[2,8]. This led to the hypothesis that the posterior epiblast and/or primitive streak/mesoderm may be the precursor to human PGCs (Model 1, Fig. 1a)[6,8]. This hypothesis was initially surprising, as a hallmark of PGC development in vivo is "repression of somatic genes" (including primitive streak/mesodermal markers[23]). However, shared transcription factors are required for both primitive streak and PGC specification[10,11,24], hinting at their intertwined origins. Alternatively, PGCs and primitive streak may be two completely independent lineages that arise from nearby precursors (Model 2, Fig. 1a). Finally, studies of cynomolgus monkey embryos have instead proposed that PGCs arise from the dorsal amnion[25].

Guided by these insights, prevailing strategies to differentiate hPSCs into PGCLCs are divided into two phases: first, exposure to primitive streak/mesoderm-inducing signals (TGFβ and WNT) for 12-48 h in monolayer cultures, followed by 3D aggregation and treatment with PGC-specifying signals (BMP, SCF, EGF, and LIF) for multiple days[6,8,14]. This highlighted key inductive signals for PGCLC specification, although the precise temporal dynamics with which key signals (e.g., WNT) are needed remains unclear. The transcriptional network that incipiently specifies human PGCs from pluripotent cells also requires further definition. Pluripotent cells and PGCs share the expression of pluripotency transcription factors[23,26]. It is currently proposed that when pluripotent cells differentiate towards PGCs, the pluripotency network is first abolished in differentiating cells and then reactivated in PGCs, at least in mice[23,26]. An alternate model is that pluripotency genes are continuously expressed from the pluripotent epiblast to the incipient PGCs, without an intermediate step where pluripotency genes are silenced. Preliminary evidence for this latter model was provided in cynomolgus monkey embryos[25] but it is unclear whether the same holds true for human PGC specification. Taken together, multiple outstanding questions continue to surround where, when, and how human PGCs are specified.

Here we develop a simplified 2D platform to generate human PGCLCs within 3.5 days of hPSC differentiation, and we demonstrate the applications of this simplified differentiation approach to provide insights into PGCLC specification. Prevailing approaches to generate human PGCLCs entail 3D aggregates treated with high concentrations of growth factors[6–8]. While 3D aggregates may be beneficial as they concentrate intercellular interactions important for lineage specification, it is challenging to direct hPSC differentiation within 3D aggregates due to multiple difficult-to-control variables, including limited diffusion of extracellular signals deep into aggregates[27]. First, we adapt human PGCLC differentiation to monolayer culture and demonstrate that temporally dynamic activation, followed by repression, of WNT signaling is critical. Transcriptomic analyses of WNT modulation at different timepoints during PGCLC differentiation demonstrate that later-stage WNT inhibition is essential for efficient PGCLC induction. This method can be robustly extrapolated to a wide range of hESC and hiPSC lines to achieve PGCLC specification in monolayer culture. Second, we also describe cell-surface markers for human PGCLCs (namely, CXCR4+PDGFRA−GARP−) that enable their ready purification via fluorescence-activated cell sorting (FACS). Third, we provide a detailed characterization of cells en route to PGCLCs specification by single-cell RNA-sequencing and find that pluripotency transcription factors OCT4 and NANOG are continuously expressed during this process, thus bridging pluripotency with germline states. We further demonstrate the significance of continued NANOG expression during in vitro PGCLC specification by immunofluorescence, live imaging, and NANOG siRNA knockdown. Finally, single-cell RNA-sequencing shows that monolayer culture-induced PGCLCs share transcriptional similarities with PGCs obtained from the human fetus as well as PGCLCs generated with currently published 3D differentiation methods.

## Results

### Temporally dynamic WNT activation, followed by inhibition, increases efficiency of human PGCLC specification

We hypothesized that temporal control over WNT signaling might be crucial for PGCLC specification, as the precise duration of WNT activity is of paramount importance to specify multiple cell types from hPSCs[28,29]. Prevailing methods for hPSC differentiation into PGCLCs generally entail two steps. First, exposure to posteriorizing signals (including TGFβ, WNT, and non-specific ROCK inhibitor Y-27632) that induce primitive streak/mesoderm for 12-60 h[6,8,14]. Second, cells are then aggregated in 3D and treated with high concentrations of BMP, EGF, SCF, LIF, and Y-27632 for multiple days to generate PGCLCs[6,8]. Using these published protocols as a framework (Fig. S1a)[6,8] and NANOS3-mCherry hPSCs to quantify the percentage of NANOS3+ PGCLCs[7], we sought to examine the temporal dynamics of WNT signaling and to generate human PGCLCs in monolayer cultures.

First, we found that in the first phase of differentiation, 12 h of exposure to posteriorizing signals (including WNT agonist CHIR99021) was optimal, in order for NANOS3-mCherry+ PGCLCs to subsequently arise at the second stage of differentiation (Fig. 1b) in monolayer cultures. We further confirmed that 12 h of posteriorizing signals was optimal across 3 additional hPSC lines (Fig. S1b), thereby reaffirming findings that 12 h treatment with posteriorizing signals is ideal for subsequent PGCLC differentiation[8]. In our hands, prolonged exposure to posteriorizing signals for 24 h−which we and others have shown generate primitive streak (PS) cells capable of subsequent endoderm and mesoderm differentiation[8,28,29]−abrogated the subsequent generation of PGCLCs in the second phase (Fig. 1b, Fig. S1b).

Subsequently, in the second phase of differentiation, we found that explicit inhibition of WNT signaling (using XAV939[30]) led to a ~2-3-fold improvement in PGCLC specification (Fig. 1c), compare "base condition" vs. "XAV939". Conversely, continued WNT activation with CHIR99021 in the second phase of differentiation completely repressed PGCLC specification (Fig. 1c). The explicit requirement for WNT inhibition (beyond simply withholding exogenous WNT) implies that differentiating hPSCs endogenously produce WNT signals[28,29,31], which inhibit PGCLC formation. This emphasizes the need to control endogenous signals to guide efficient differentiation and is consistent with how PRDM14 inhibits endogenous WNT signaling during PGCLC specification[13].

Given the importance of this initial 12-h WNT pulse in the first phase of differentiation, we sought to molecularly detail the differentiated cells at day 0.5 (D0.5), which constitute a key intermediate en route to PGCLC differentiation. Single-cell RNA-sequencing (scRNA-seq, using the 10X Genomics droplet-based platform[32]) revealed that these hPSC-derived D0.5 cells were fairly homogenous, and continued to highly express pluripotency transcription factors OCT4 and NANOG, although SOX2 decreased (Fig. 1d, Fig. S1c, d). D0.5 cells also began to concomitantly express posterior epiblast/future primitive streak markers such as MIXL1, BRACHYURY, FGF8, and NODAL (Fig. 1e, Fig. S1c, d). However, D0.5 cells generally expressed posterior epiblast/primitive streak markers at lower levels (apart from FGF8) compared to D1 primitive streak cells that were generated by 24 h of exposure to posteriorizing signals (Fig. 1e). Consistent with the use of TGFβ and WNT agonists to induce D0.5 cells, these cells demonstrated an active transcriptional response to both signaling pathways, including TGFβ target genes (FOLLISTATIN, ID1 and LEFTY2) and WNT target genes (SP5) (Fig. S1c, d). The D0.5 cell population did not show substantial transcriptional heterogeneity, as shown by scRNA-seq (Fig. S1e).

We provisionally designate these intermediate cells generated upon 12-h exposure to posteriorizing signals as "posterior epiblast" to distinguish them from primitive streak. As discussed above, D1 PS cells can generate endoderm and mesoderm, but not PGCLCs (Fig. 1b, Fig. S1b). We propose that D0.5 cells correspond to "posterior epiblast" based on how, in mouse embryos, the post-implantation pluripotent

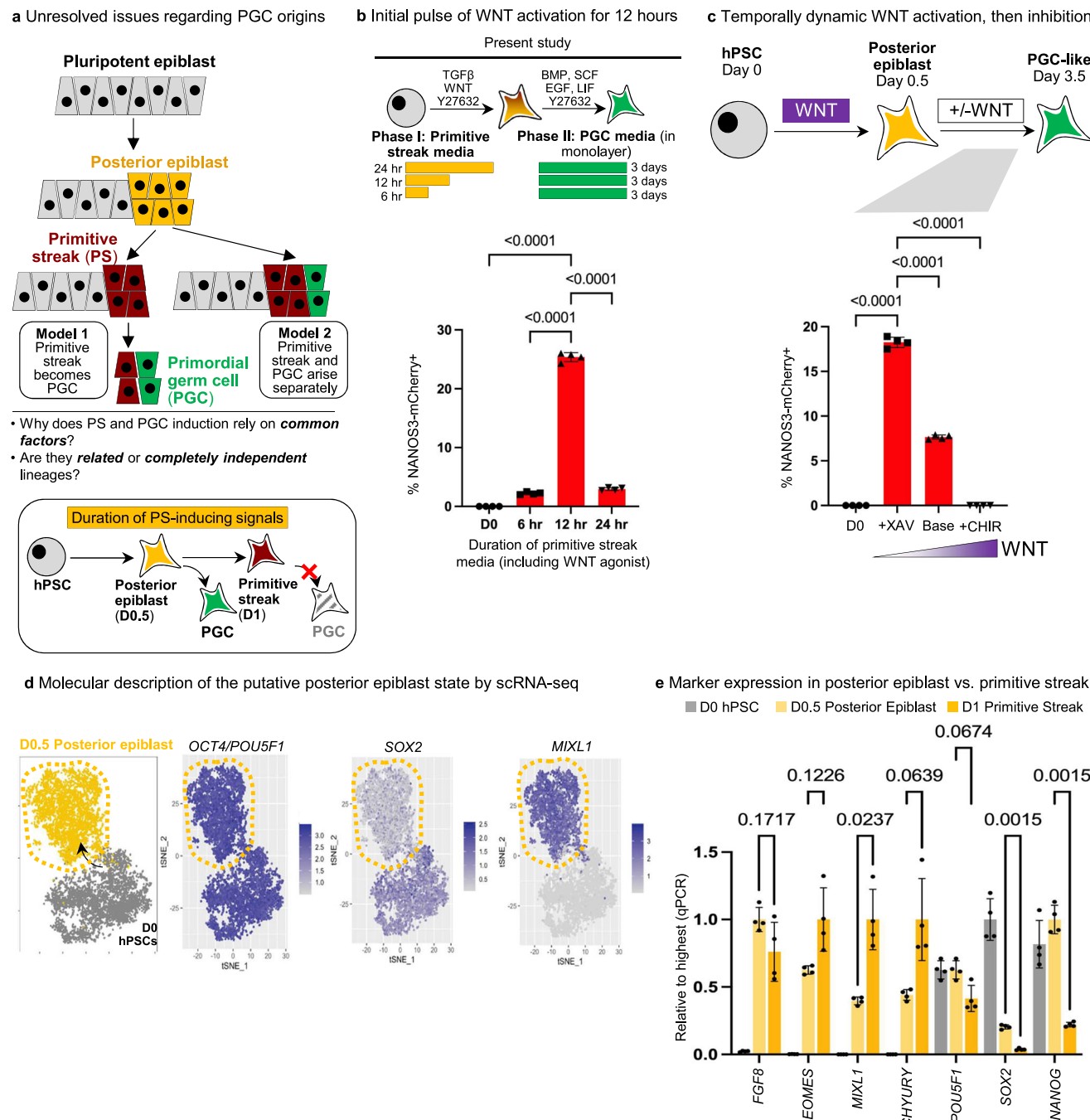

**Fig. 1 | Temporally dynamic WNT activation, followed by inhibition, promotes human PGCLC formation. a** Schematic of proposed steps of PGC development in early embryogenesis and biological questions. In Model 1, primitive streak/meso-derm-like cells give rise to PGCs (Sasaki et al.[6]). In Model 2, primitive streak and PGCs arise separately. Depicted cell positions are based on pig embryos (Kobayashi et al.[8]). **b** Exposure to primitive streak-inducing signals for 12 h is optimal for subsequent PGCLC specification; *NANOS3-mCherry* hESCs were exposed to primitive streak-inducing signals (Activin + CHIR + Y-27632) for either 6, 12, or 48 h (phase I), and then transferred to PGCLC-specifying media for 3 days (phase II), and flow cytometry was then performed. Source data are provided as a Source Data file. **c** WNT inhibition promotes differentiation of posterior epiblast into PGCLCs; *NANOS3-mCherry* hESCs were differentiated into posterior epiblast

using a WNT agonist (12 h, phase I), and then were transferred into PGCLC-specifying media for 3 days in the presence of WNT agonist (CHIR99021) or WNT inhibitor (XAV939) (phase II); flow cytometry was then performed on D3.5. Source data are provided as a Source Data file. **d** t-SNE plot of single-cell RNA-sequencing of posterior epiblast cells (D0.5 of differentiation) and hPSCs (D0 of differentiation) showing expression of pluripotency markers *OCT4/POU5F1* and *SOX2* and primitive streak marker *MIXL1*. **e** qPCR analysis of D0 (hPSC), D0.5 (posterior epiblast), and D1 (primitive streak) of differentiation showing expression of pluripotency and pri-mitive streak markers; qPCR data were normalized to the sample with the highest expression (which was set = 1.0). Statistical test−Two-Way ANOVA. *n* = 4 biological replicates/group for all. *P* values are shown above bars, comparing D0.5 and D1; error bars = standard error of mean. Source data are provided as a Source Data file.

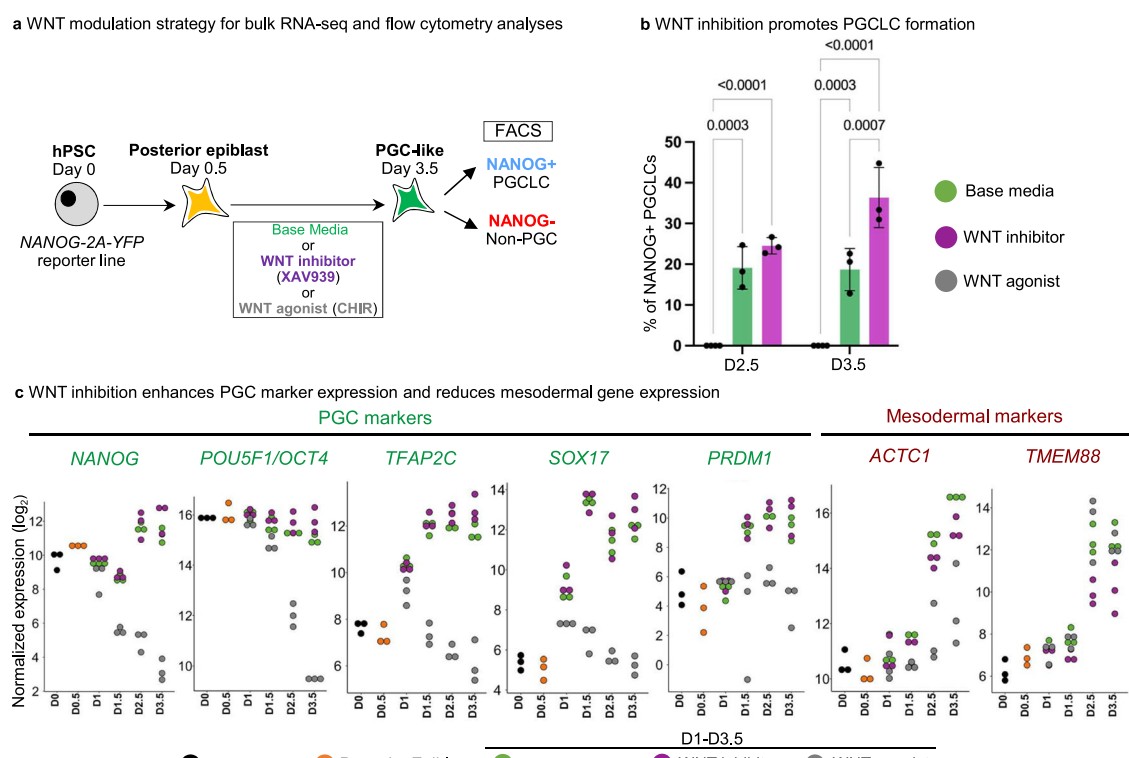

**Fig. 2 | Subsequent WNT inhibition promotes PGCLC generation and represses mesodermal markers. a** Schematic of WNT agonism (CHIR) or WNT antagonism (XAV939) or no WNT manipulation (base media) at different timepoints during PGCLC monolayer differentiation. **b** FACS data showing efficiency of generating NANOG+ PGCLCs under different conditions of WNT pathway manipulation at Day 2.5 and Day 3.5. Data are presented as mean values ± SEM. Statistical test–two-way ANOVA, with Tukey multiple testing correction. *n* = 3 biological replicates/group for all except for CHIR99021 where *n* = 4 biological replicates/group. Adjusted *P* values are shown above error bars. Source data are provided as a Source Data file. **c** Log2 normalized expression levels of PGC and mesodermal markers in Base media vs. XAV939 vs. CHIR treated samples at different timepoints. *n* = 3 biological replicates/group.

epiblast is formed by embryonic day 5.5 (E5.5), but then PS markers (e.g., Brachyury) are transiently expressed in the posterior region of the epiblast (~E6-E6.25) immediately *prior* to overt formation of the morphologically-conspicuous PS (~E6.5)[22]; similar results have been reported in pig embryos[8]. However, we note that early human post-implantation embryos remain inaccessible for analysis, and thus assignment of terms such as "posterior epiblast" or "primitive streak" in human is premised on evolutionary homology to other mammals such as pig and mouse[27]. In summary, this discloses a unique transcriptional signature for D0.5 posterior epiblast cells, wherein pluripotency factors *OCT4* and *NANOG* are co-expressed together with primitive streak markers.

## Subsequent WNT inhibition promotes PGCLC specification and represses mesodermal genes

To further investigate the role of WNT inhibition in the second phase of differentiation, we added WNT agonist (CHIR99021) or WNT inhibitor (XAV939) to D0.5 posterior epiblast cells for the remainder of differentiation and performed bulk transcriptomic analyses at different timepoints (Fig. 2a, Fig. S2a–f, Supplementary Data 5).

WNT inhibition promoted PGCLC formation (Fig. 2b), induced higher levels of PGC markers (*NANOG, POU5F1, TFAP2C, SOX17* and *PRDM1*) and repressed mesodermal genes (*ACTC1* and *TMEM88*) (Fig. 2c). While lack of any WNT inhibition ("base media" alone) still gave rise to PGCLCs, it did so less efficiently than with WNT inhibition (Fig. 2b). WNT ligands (e.g., *WNT5B*) and WNT target genes (e.g., *LEF1* and *SP5*) were upregulated in PGCLCs generated from the standard "base media" condition, but their expression was repressed by WNT inhibitor treatment (Fig. 3a–d, Fig. S2f, Supplementary Data 5). Indeed, quantifying the total expression of all known WNT ligands and WNT

target genes revealed that WNT inhibition repressed overall levels of endogenous WNT ligand expression and WNT pathway activation (Fig. 3a, b). Continued WNT activation instead induced primitive streak markers and strongly repressed PGCLC formation (Figs. S3a, b, 2c).

We conclude that temporally dynamic WNT activation, followed by inhibition, enhances human PGCLC specification by repressing endogenously activated WNT ligands, thus providing an additional dimension to our knowledge of PGCLC development. This parallels how WNT is initially required, and then is dispensable, for pig PGC specification in embryonic explant cultures[8].

## Generation of human PGCLCs in monolayer conditions

After generating presumptive posterior epiblast cells, we tested whether continuous BMP, SCF, LIF, and EGF activation[6–8] was required for the entire second phase of PGCLC differentiation in monolayers. First, omitting BMP4 from the culture media from D1.5-D2.5 led to a ~2.5-fold increase in PGCLC specification, while the absence of SCF and EGF from D0.5-D1.5 was superfluous (Fig. S4a). Second, past 3D differentiation methods used high BMP4 concentrations (200–500 ng/mL)[6–8], but in our monolayer conditions, significantly lower (25-fold lower) BMP4 concentrations were needed (Fig. S4b). This is consistent with the notion that BMP signaling does not effectively act across large hPSC clusters[33,34]. 3D aggregates may therefore impair BMP signaling, thus emphasizing potential benefits of a monolayer differentiation system. Third, LIF, which is commonly used to enhance PGCLC survival[6–8], was dispensable in our platform (Fig. S4c). Fourth, we observed a peak of PGCLC formation by day 3.5 of differentiation (Fig. S4d).

Combining these improvements together, we developed a monolayer, serum-free protocol (Fig. 4a) to generate consistently and reproducibly 20-30% pure NANOS3-mCherry⁺ PGCLCs within

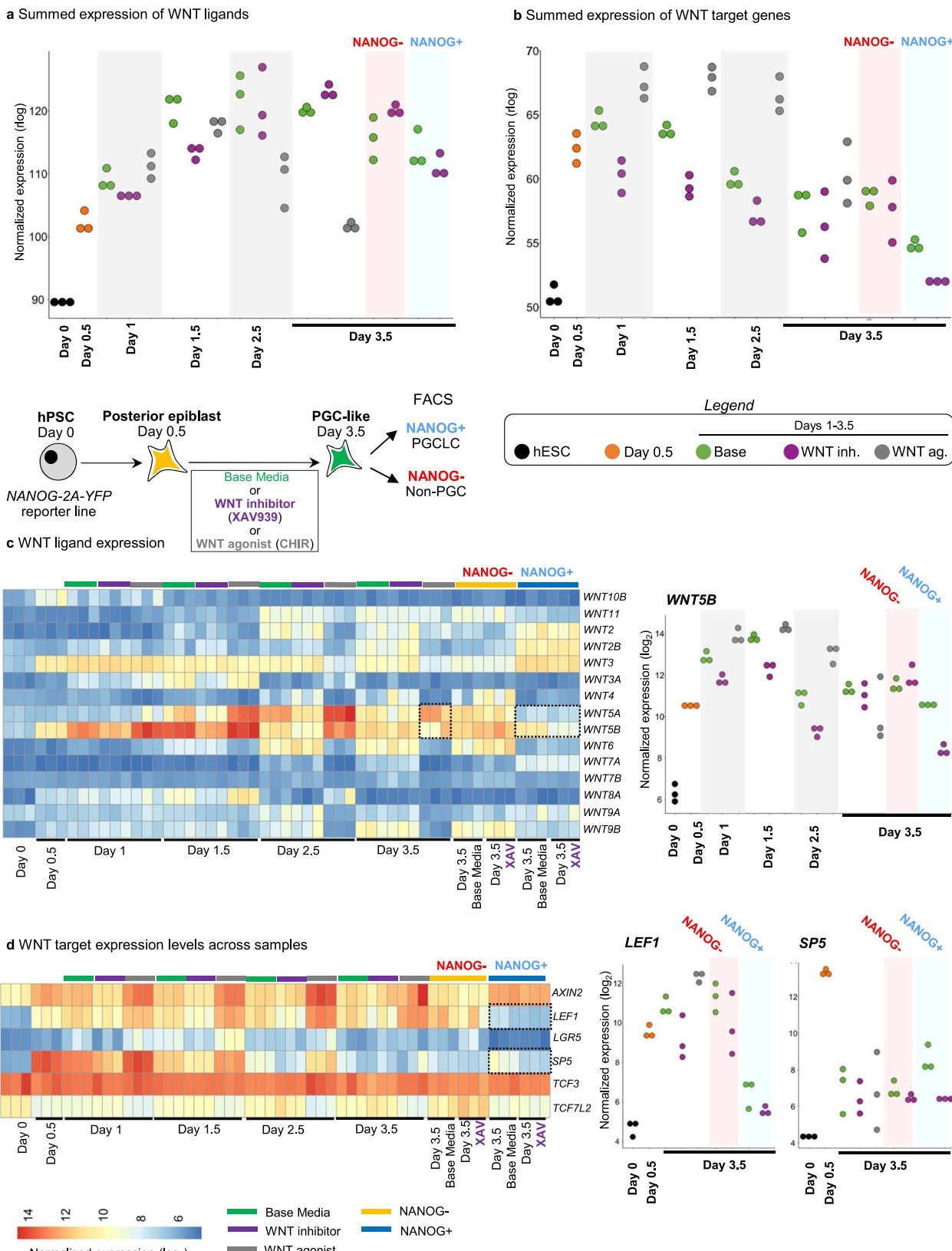

**Fig. 3 | Subsequent WNT inhibition suppresses the expression of endogenous WNT pathway ligands and target genes. a** Summed expression levels of all WNT ligands in all the samples. List of expressed WNT ligands are shown in **c**. *n* = 3 biological replicates/group. **b** Summed expression levels of WNT targets in all the samples. List of selected WNT targets are shown in **d**. **c** Left: Heatmap of all expressed WNT ligands across all samples. Right: endogenous *WNT5B* expression in all samples across different timepoints. **d** Left: Heatmap of all selected known WNT targets across all samples. Right: endogenous *LEF1* and *SP5* expression in all samples across different timepoints.

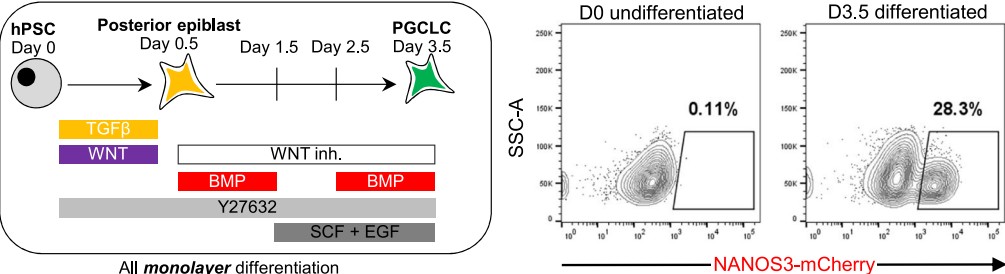

**a** New signaling strategy: monolayer PGCLC induction

**b** Generation of NANOS3+ PGCs in monolayer conditions

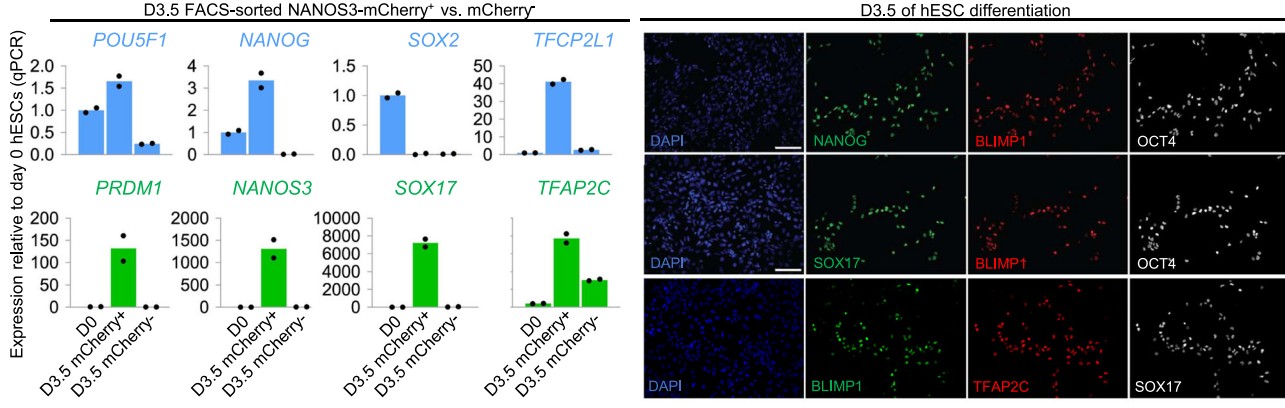

**c** hPSC-derived NANOS3+ PGCs express PGC markers

**d** hPSC-derived D3.5 differentiated populations express PGC marker proteins

**e** Quantification of PGCLC differentiation across hPSC lines

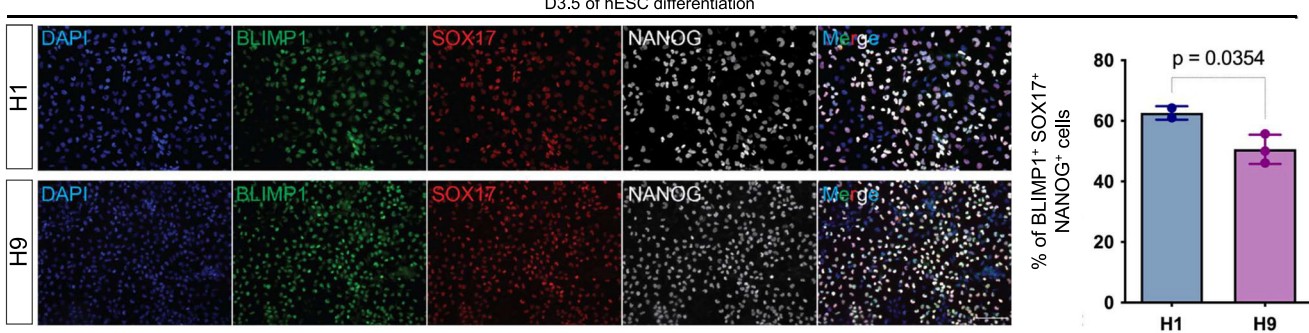

**Fig. 4 | A simplified monolayer platform to generate human PGCLCs.**
**a** Schematic of the 2D monolayer PGCLC differentiation protocol reported in this manuscript. **b** Flow cytometry analysis of *NANOS3-mCherry* hESC shows fluorescent reporter expression before or after 3.5 days of differentiation. **c** qPCR analysis of NANOS3-mCherry+ PGCLCs and NANOS3-mCherry- non-PGCLCs derived after 3.5 days of differentiation, as shown in **h**; as a negative control, undifferentiated hPSCs (D0) were also analyzed, and gene expression is shown relative to undifferentiated hPSCs (which was set = 1.0). Data are presented as mean values. Source data are provided as a Source Data file. **d** Immunostaining of hPSCs differentiated for D3.5 showing expression of PGC markers in a subset of cells (nuclear counterstain: DAPI). Scale bar = 100 μm. **e** Validation of protein expression in D3.5 human ESCs. Each panel indicates the corresponding marker. The graph on the right represents the quantification of triple positive cells at D3.5 in the differentiation protocol. See Materials and Methods for details on quantification method. Each column represents mean with SEM for at least two biological replicates. $n = 19,687$ for H1, $n = 57,067$ for H9. DAPI was used as nuclear counterstain. Representative of two independent experiments. *P* values are shown above bars; error bars = standard error of mean. Statistical test–unpaired *t* test with Welch's correction. Source data are provided as a Source Data file.

3.5 days of in vitro differentiation (Fig. 4b). NANOS3-mCherry+ PGCLCs purified by fluorescence-activated cell sorting (FACS) expressed hallmark PGC markers, including *POU5F1* (*OCT4*), *NANOG*, *TFCP2L1*, *PRDM1* (*BLIMP1*), *NANOS3* and *TFAP2C* (*AP2γ*) (Fig. 4c). At the protein level, PGCLCs co-expressed NANOG, PRDM1/BLIMP1, SOX17 and OCT4/POU5F1 (Fig. 4d, e). Additionally, PGCLCs generated through our protocol contained 5-hydroxymethylcytosine (Fig. S4e), an important intermediate in DNA demethylation and thus the epigenetic resetting of PGCs/PGCLCs[7,35].

Finally, we independently validated this monolayer differentiation protocol using a separate *SOX17-GFP* knock-in reporter hPSC line[36] to track the expression of human PGCLC marker SOX17[7] (Fig. S4f, g). We ultimately applied our differentiation protocol across 5 additional hESC/hiPSC lines and found that it reproducibly generated PGCLCs (detailed below; Fig. 4e, Fig. 5, Fig. S5).

### Surface-marker profile of hPSC-derived PGCLCs: CXCR4+ PDGFRα− GARP−

Current protocols to generate human PGCLCs in monolayers (this study) or aggregates[6–8] generate heterogeneous cell populations containing a subset of PGCLCs; therefore cell-surface markers to selectively identify and purify PGCLCs would be a boon. EPCAM, ITGA6, PDPN, CD38, KIT, and alkaline phosphatase activity have been previously reported to enrich for human or non-human primate

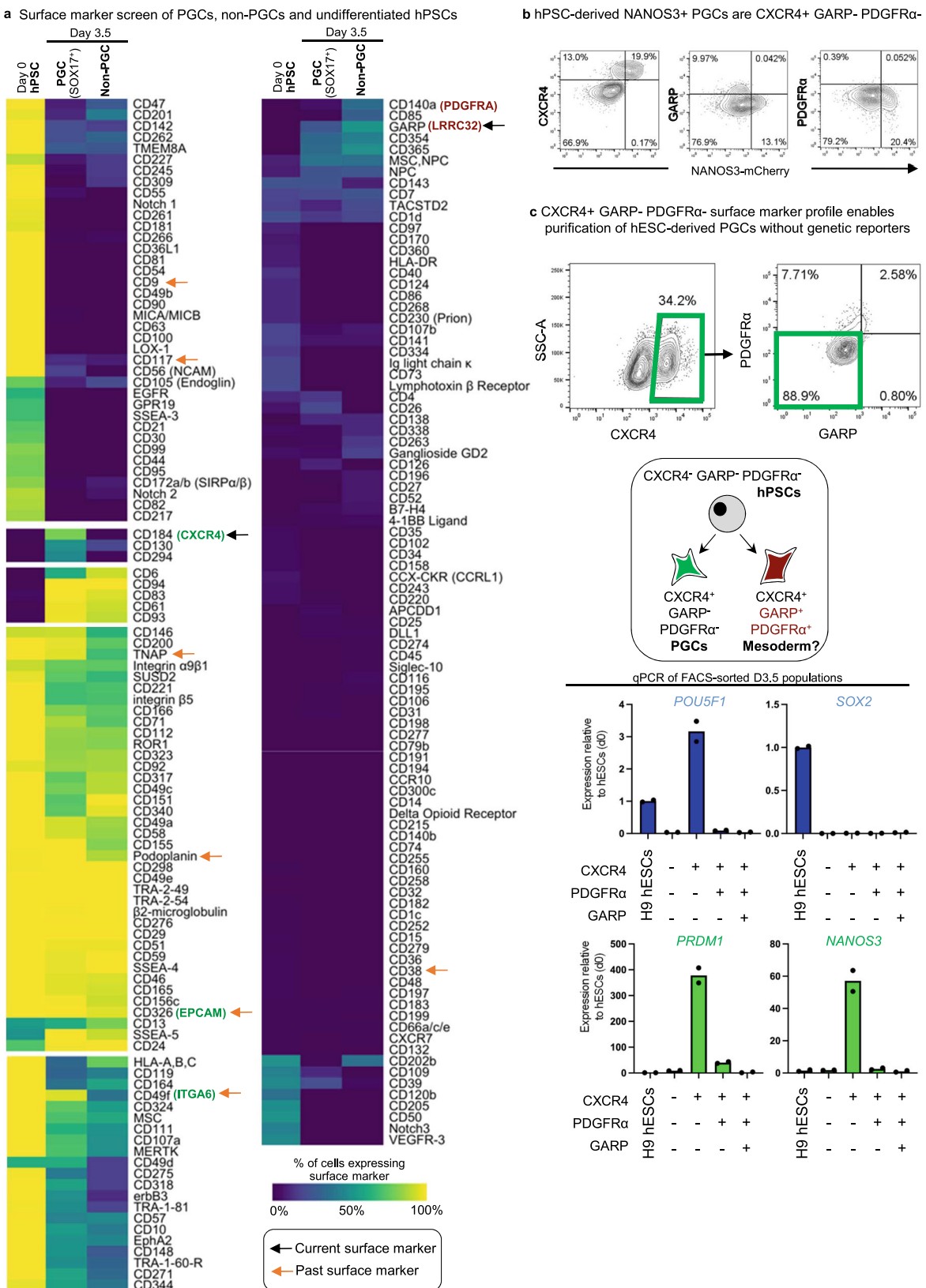

**a** Surface marker screen of PGCs, non-PGCs and undifferentiated hPSCs

**b** hPSC-derived NANOS3+ PGCs are CXCR4+ GARP- PDGFRα-

**c** CXCR4+ GARP- PDGFRα- surface marker profile enables purification of hESC-derived PGCs without genetic reporters

qPCR of FACS-sorted D3.5 populations

PGCLCs[6,7,9]. However, at the transcriptional level, many of these markers are also expressed on undifferentiated hPSCs (Fig. S5a), consistent with past reports that hPSCs express these marker proteins[6,7,9]. Using our optimized monolayer platform for PGCLC differentiation, we thus sought to discover alternative cell-surface markers to purify PGCLCs.

We robotically screened the expression of 369 cell-surface markers using high-throughput FACS[28] on *SOX17-GFP* hPSCs differentiated into D3.5 SOX17-GFP+ PGCLCs vs. SOX17-GFP− non-PGCLCs; undifferentiated hPSCs were also included as a negative control (Fig. 5a). This confirmed that EPCAM, ITGA6, PDPN and alkaline phosphatase[6,7,9]

**Fig. 5 | High-throughput screening identifies a CXCR4+ PDGFRα⁻ GARP⁻ cell-surface marker signature for hPSC-derived PGCLCs. a** Heatmap of surface markers expressed in undifferentiated hPSC (D0), D3.5 SOX17-GFP⁺ PGCLCs, and D3.5 SOX17-GFP⁻ non-PGCLCs identified from LEGENDScreen; to discriminate PGCLCs vs. non-PGCLCs, *SOX17-GFP* hESCs were differentiated for D3.5 and then subgated on GFP⁺ and GFP⁻ before further analysis of surface marker expression; color shades represent the percentage of cells in each expression that are positive for a given marker; each row depicts expression of a single surface marker across all populations. **b** Flow cytometry analysis of D3.5 differentiated *NANOS3-mCherry*

hESCs reveals CXCR4, GARP, and PDGFRα expression relative to NANOS3-mCherry fluorescent reporter expression. **c** Flow cytometry gating strategy to identify CXCR4⁺/GARP⁻/PDGFRα⁻ PGCLCs derived from H9 hESCs (that did not carry any fluorescent reporters) that were differentiated for D3.5; various cell populations from the D3.5 population were FACS sorted and subject to qPCR analysis, revealing that pluripotency and PGC markers are restricted to the CXCR4⁺/GARP⁻/PDGFRα⁻ subset and therefore reaffirming its PGCLC identity. *N* = 2 biological replicates, Data are presented as mean values. Source data are provided as a Source Data file.

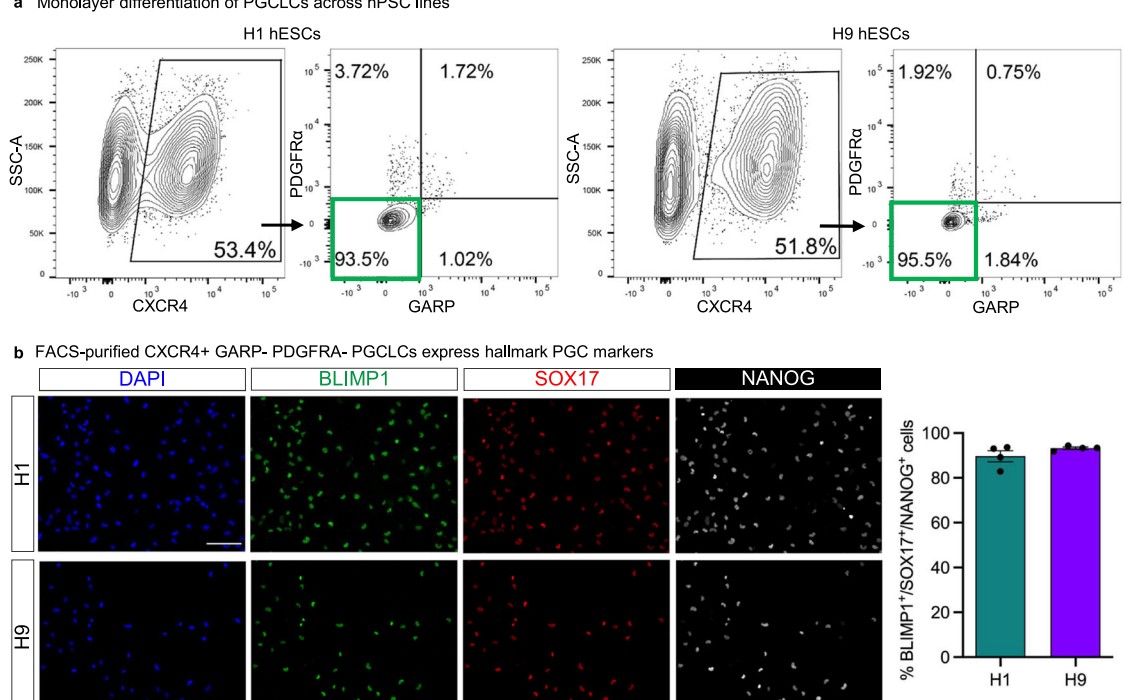

**Fig. 6 | Validation of 2D hPGCLC induction protocol. a** Representative FACS plots showing gating strategy based on CXCR4, PDGFRa, and GARP signals. Briefly, cells were first gated based on CXCR4 signal (left panels for both cell lines). These cells were then further analyzed to exclude PDGFRα⁺ and GARP⁺ cells, highlighted by the green rectangle. Source data are provided as a Source Data file. **b** Representative images of D3.5 hPGCLCs validated by immunofluorescence staining following FACS

purification from three independent experiments. The markers used for validation are indicated on each panel. Scale bar = 100 μm. Quantification based on manual counting of triple positive cells for the indicated markers. Each column represents the mean with SEM for four biological replicates. *n* = 1237 for H1, *n* = 736 for H9. DAPI was used as nuclear counterstain. Source data are provided as a Source Data file.

were not specific markers since they were both expressed on hPSCs as well as PGCLCs (Fig. 5a).

In our analysis, the most specific positive marker for SOX17-GFP⁺ PGCLCs was the chemokine receptor CXCR4/CD184 (Fig. 5a), which similarly marked NANOS3-mCherry⁺ PGCLCs (Fig. 5b). Intriguingly, in model organisms, CXCR4 is known to be expressed by PGCs, and enables PGC migration towards the gonads in response to CXCL12[37–40]; this may also be conserved in human[12]. However, CXCR4 is also expressed on mesodermal derivatives[41], and therefore negative expression of mesodermal markers is necessary to exclude mesoderm. We found that the mesodermal markers PDGFRα/CD140A[42] and GARP/LRRC32[28] were expressed on the D3.5 non-PGCLCs (Fig. 5a, b), thus providing a means to eliminate mesoderm. At the transcriptional level, RNA-seq reaffirmed that PGCLCs were *CXCR4 + PDGFRA−*, in contrast to commonly used PGCLC markers *EPCAM* and *ITGA6*[6,9], which were both expressed on PGCLCs and hPSCs and were therefore less specific (Fig. S2d).

Taken together, by relying on a combination of positive (CXCR4) and negative (PDGFRα, GARP) markers, we defined a CXCR4⁺ PDGFRα⁻ GARP⁻ surface marker profile for hPSC-derived

PGCLCs. Logical combinations of positive and negative markers have likewise proven decisive in the purification of specific cell-types in blood and other tissues[43]. In differentiated D3.5 cultures, the CXCR4⁺ PDGFRα⁻ GARP⁻ fraction contained all PGCLCs; other combinations of these surface markers did not enrich for PGCLCs (Fig. 5c).

## PGCLCs can be consistently generated across diverse hESC and hiPSC lines

We validated our monolayer differentiation protocol as well as the CXCR4⁺ PDGFRα⁻ GARP⁻ sorting strategy across an additional panel of five wildtype hESC and hiPSC lines (encompassing both male and female lines) that did not bear knock-in reporters. The present monolayer differentiation method generated an average of 46.3 ± 8.5% pure CXCR4⁺ PDGFRα⁻ GARP⁻ PGCLCs (Fig. 6a, Fig. S5b). Our CXCR4⁺ PDGFRα⁻ GARP⁻ cell-surface marker signature allowed us to purify differentiated PGCLCs across all hESC and hiPSC lines tested, using our improved differentiation strategy and without recourse to transgenic reporters (Fig. 6a, Fig. S5b). Across all lines, FACS purification of CXCR4⁺ PDGFRα⁻ GARP⁻ PGCLCs enriched the expression of hallmark

PGC markers (Fig. S5c). Immunostaining of FACS-purified CXCR4+ PDGFRα− GARP− PGCLCs showed that most FACS-sorted cells (89.7 ± 2.5% for H1, 93.5±0.5% for H9) co-expressed PGC hallmark proteins BLIMP1/PRDM1, SOX17, and NANOG (Fig. 6b, Fig. S5d). This exemplifies the fidelity of PGCLC specification across distinct genetic backgrounds and demonstrates the utility of the CXCR4+ PDGFRα− GARP− surface marker profile.

### Tracking the trajectory and uniformity of PGCLC specification in vitro using single-cell RNA-sequencing

Next, we illuminated the stepwise changes in gene expression as hPSCs incipiently differentiated into posterior epiblast (D0.5) and then into PGCLC-containing populations (D3.5) by performing scRNA-seq[32] of all these populations (Fig. 7a). scRNA-seq was important to detail the cellular diversity of this population and to obtain a refined and specific transcriptional signature only for the PGCLCs. As a negative control, we also performed scRNA-seq of D2 definitive endoderm[29]—a lineage derived from the PS (and thus, on a related but distinct lineage path from PGCs)—to clarify the relationship between human PGCs and endoderm, given that human PGCLCs reportedly express "endodermal" marker SOX17[6,7]. Taken together, we analyzed 24,473 cells by scRNA-seq, with a median of >4000 genes detected per cell in each cell population (Fig. S6a).

scRNA-seq showed that the D3.5 bulk differentiated population was transcriptionally heterogeneous, comprising two major subsets (Fig. 7b). One subset comprised PGCLCs expressing NANOS3, TFAP2C, and KLF4 (Fig. 7b, Fig. S6c). Intriguingly, the non-PGCLCs expressed lateral mesoderm marker HAND1 and the cardiac mesoderm markers TMEM88, MYL4, and ACTC1[28,44] (Fig. 7b, e, Fig. S8c). This suggests that the "mis-differentiated", non-PGCLCs at D3.5 are mesoderm-like cells, as evinced by HAND1 protein expression in the D3.5 non-PGCs (Fig. 7c). Indeed, the principal signals we used to differentiate posterior epiblast into PGCLCs (BMP activation and WNT inhibition) are the same ones that differentiate primitive streak into cardiac mesoderm[28], suggesting that some cells on the wrong differentiation trajectory respond to these same signals to adopt mesoderm-like identity. Pseudotemporal ordering of cells[45] from Day 0, Day 0.5, and Day 3.5 delineated two main trajectories, with one main branch leading to PGCLCs and another to the "mis-differentiated", mesoderm-like non-PGCLCs (Fig. 7d, Fig. S7a, b). Key PGC markers NANOS3, NANOG, and TFAP2C were upregulated in the trajectory leading to PGCLCs, but not in the non-PGCLC trajectory (Fig. S7c). BMP4, IGF2, LEF1, and TCF4 were instead upregulated in the non-PGCLCs (Fig. S7c).

Integrated scRNA-seq analysis of all populations revealed the stepwise changes in gene expression as pluripotent cells segue into D0.5 posterior epiblast and, finally, D3.5 PGCLCs (Fig. 7, Fig. S7). Posterior epiblast markers BRACHYURY, MIXL1 and NODAL were transiently expressed at D0.5 (consistent with how posterior epiblast/ primitive streak transcription factors are required for mammalian PGC specification[10,11,24]), but were subsequently downregulated in D3.5 PGCLCs (Fig. 7e). This is consistent with the observed "repression of somatic genes" in fully-formed PGCLCs[23], although we note that these genes are nonetheless briefly expressed in their precursors (the posterior epiblast). Of note, D3.5 PGCLCs generated in our system did not express BRACHYURY (Fig. 7e), which is expressed by PGCLCs generated by other differentiation systems[6,7,46]. This may be explained by our inhibition of WNT signaling, as WNT is known to directly upregulate BRACHYURY expression[47].

Our side-by-side comparison of PGCLCs and endoderm confirmed that they shared common markers SOX17 and PRDM1[6,7]; however, D3.5 PGCLCs expressed multiple unique markers that were not found in endoderm, including NANOG, NANOS3, TFAP2C, KLF4, and TCL1B (Fig. 7e, Fig. S7c), thus disclosing a single-cell transcriptional signature for hPSC-derived PGCLCs.

### CXCR4+ PDGFRα− GARP− cells are transcriptionally highly enriched for PGCLCs

To overcome the population heterogeneity evident at D3.5 of differentiation (Fig. 7b and Fig. S8), we asked whether our cell-surface markers (CXCR4+ PDGFRα− GARP−) could enable the purification of nearly homogeneous PGCLCs. While past combinations of cell-surface markers could isolate PGCLCs that were enriched for NANOS3, PRDM1, and TFAP2C expression[6,7], we surmised that single-cell RNA-seq of FACS-sorted PGCLCs would rigorously assess whether they were truly homogeneous at the transcriptome-wide level. scRNA-seq of FACS-sorted CXCR4+ PDGFRα− GARP− D3.5 PGCLCs revealed four subsets: three subsets comprised PGCLCs, cumulatively accounting for 97.2% of the total population (Fig. 7f, Fig. S8a−c). These three PGCLC subsets expressed similar levels of archetypic PGC markers (e.g., NANOG), but could be distinguished by cell cycle genes and higher expression of TFAP2A and EDN1 in a small subset of PGCLCs (Fig. S8a−d). The remaining 2.8% of cells were PDGFRα+ mesoderm-like cells, likely owing to imperfect FACS sorting for CXCR4+ PDGFRα− GARP− cells (Fig. 7f, Fig. S8a−d). This result thus reaffirms the power of our cell-surface marker profile to precisely isolate PGCLCs from a heterogeneous cell population, thus opening the door to downstream functional and molecular analyses of purified PGCLCs.

To assess if the PGCLCs derived with (XAV939) and without WNT inhibition ("base media") were transcriptionally similar, we performed bulk RNA-seq of FACS-purified NANOG + CXCR4+ PGCLCs vs. NANOG-CXCR4- non-PGCLCs obtained from both conditions. Pearson correlation analysis revealed high correlation between PGCLCs derived from both conditions, with few differentially expressed genes (Fig. 7g, Fig. S2b, c, e, f, Supplementary Data 5). Thus, although the efficiency of PGCLC differentiation differs with and without WNT inhibition, the PGCLCs obtained from both conditions were transcriptionally similar.

### NANOG is continuously expressed in the transition from pluripotency to PGCLCs

We then investigated expression of pluripotency markers during germline differentiation: a quintessential feature of early germ cells (unlike most somatic cell types) is that they express pluripotency transcription factors[23,26]. The prevailing model is that upon early differentiation, pluripotent cells initially downregulate pluripotency factors, but subsequently only cells allocated to the germline "re-express" pluripotency factors[23,26] (Fig. 8ai). By contrast, recent observations of cynomolgus macaque embryos suggested that NANOG is continuously expressed as PGCs incipiently arise from their precursors (Fig. 8ai), inferred from fixed embryos spanning different timepoints[25]. However, similar observations have been precluded in human embryos, as the pertinent developmental stages remain inaccessible.

To assess which of the two models may pertain to human PGCLCs, we computationally ordered differentiating cells in our scRNA-seq dataset along an inferred "pseudotime"[48], and observed that POU5F1 and NANOG were continuously expressed during the transition from pluripotency to posterior epiblast to hPGCLCs (Fig. 8aii, Fig. S7c). This thus implies continuous expression of pluripotency factors in the transition from pluripotency to germline fate. We sought to experimentally validate this prediction by tracking NANOG expression at the single-cell level. To this end, we engineered NANOG-2A-YFP reporter hESCs, using Cas9/AAV6 genome editing[49] to insert a 2A-YFP reporter immediately downstream of the NANOG gene without disrupting its coding sequence[50].

NANOG was continuously expressed during the hPSC-to-germline transition, without evidence for NANOG downregulation followed by re-expression (Fig. 8b, Fig. S9a). Undifferentiated hPSCs, D0.5 posterior epiblast cells, and D1.5 cells were largely NANOG+ CXCR4− (Fig. S9a). By D2.5-D3.5, a subpopulation continued to express NANOG but gained CXCR4, thus transitioning to NANOG+ CXCR4+ PGCLCs (Fig. 8b, Fig. S9a). By contrast, by D2.5-D3.5, other cells lost NANOG, thus

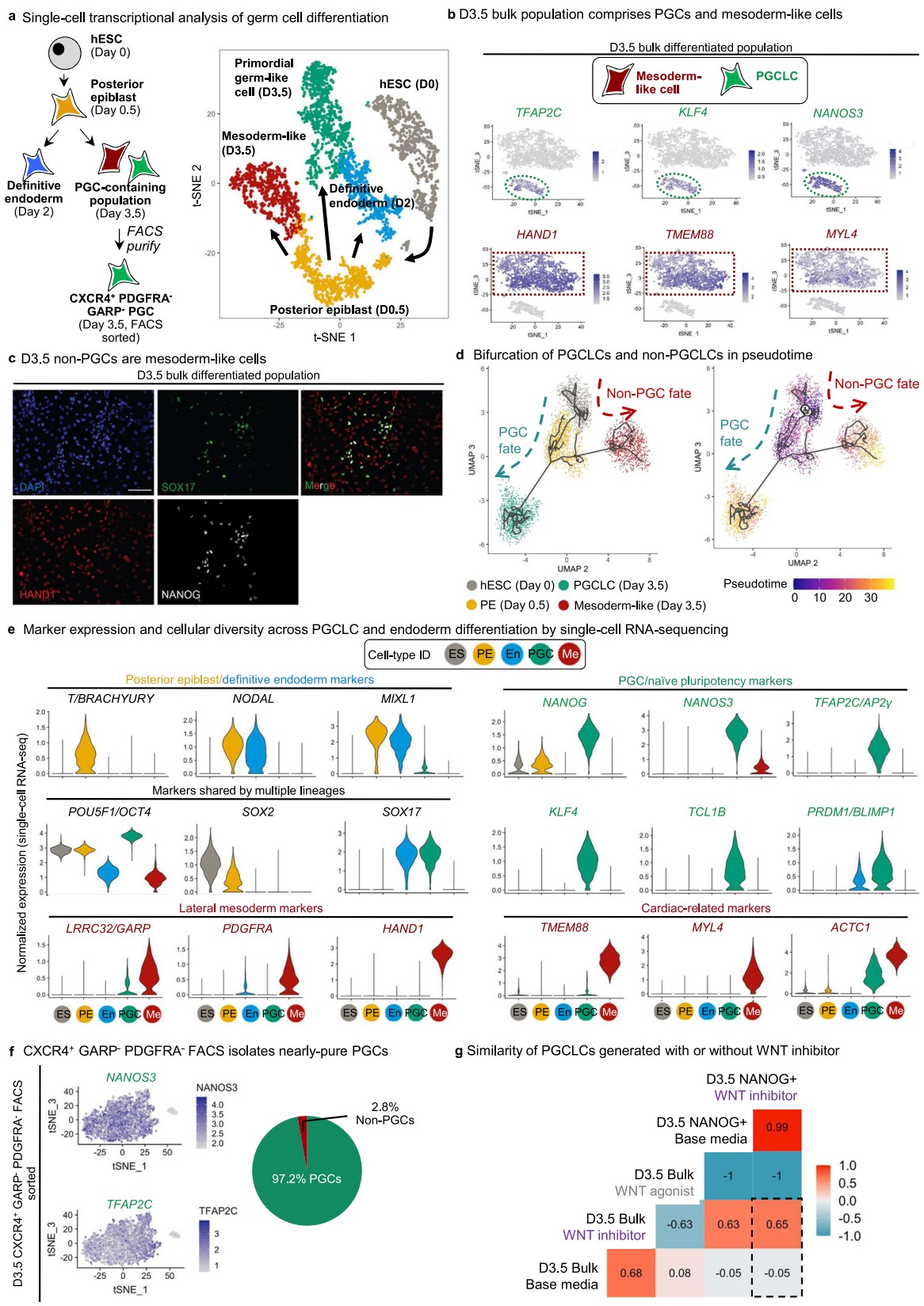

**a** Single-cell transcriptional analysis of germ cell differentiation

**b** D3.5 bulk population comprises PGCs and mesoderm-like cells

**c** D3.5 non-PGCs are mesoderm-like cells

**d** Bifurcation of PGCLCs and non-PGCLCs in pseudotime

**e** Marker expression and cellular diversity across PGCLC and endoderm differentiation by single-cell RNA-sequencing

**f** CXCR4⁺ GARP⁻ PDGFRA⁻ FACS isolates nearly-pure PGCs

**g** Similarity of PGCLCs generated with or without WNT inhibitor

differentiating into NANOG⁻ CXCR4⁻ non-PGCLCs (Fig. 8b, Fig. S9a). We independently confirmed these results, by using intracellular flow cytometry and immunostaining to directly stain for NANOG protein itself (Fig. 8c, Fig. S9b). D0 hPSCs and D0.5 posterior epiblast cells were NANOG⁺ OCT4⁺, but at D1.5, some NANOG⁺ OCT4⁺ cells began to co-express the PGC transcription factor SOX17 (Fig. 8c, Fig. S9c).

We then rigorously tested that NANOG is continuously expressed from pluripotency to germline fate through a continuous means of measurement: live imaging. Live imaging of *NANOG-2A-YFP* reporter hESCs showed that undifferentiated hPSCs were NANOG⁺, and during PGCLC differentiation, a subset of cells progressively expressed higher levels of NANOG (Fig. 8d, Movie S1).

**Fig. 7 | Single-cell RNA-sequencing reveals stepwise changes in gene expression, transcriptional trajectories, and cellular diversity during hPSC differentiation to PGCLCs. a** Schematic of stages profiled for single-cell RNA-sequencing (scRNA-seq): D0 hPSCs, D0.5 posterior epiblast, D3.5 bulk population, D3.5 FACS-sorted CXCR4+/GARP−/PDGFRα− PGCLCs and D2 definitive endoderm (*left*); *t*-SNE projection of the combined scRNA-seq data sets, where single cells are colored by their cluster annotation (*right*). **b** *t*-SNE projection of hPSC-derived D3.5 bulk population shows that it is heterogeneous and segregates into 2 major clusters: a PGCLC cluster expressing PGC markers (*TFAP2C, KLF4, NANOS3*) and mesoderm-like cells (non-PGCLCs) expressing mesoderm markers (*HAND1, TMEM88, MYL4*). **c** Immunostaining of hPSC-derived D3.5 bulk population confirms that it is heterogeneous, comprising a mixture of PGCLCs (SOX17+, NANOG+) and non-PGCLCs (HAND1+) (nuclear counterstain: DAPI). Scale bar = 100 μm. Representative images from 4 independent experiments. **d** Pseudotemporal ordering of hESCs

differentiating to PGCLCs or non-PGCLCs (mesoderm-like cells). **e** Violin plots of scRNA-seq data show expression of posterior epiblast, pluripotency, lateral mesoderm, cardiac, PGC, and naive pluripotency markers across the five different cell-types (clusters) identified from the combined scRNA-seq dataset (comprising merged D0, D0.5 posterior epiblast, D3.5 bulk, D3.5 FACS-sorted PGCLCs and definitive endoderm scRNA-seq datasets). **f** *t*-SNE projection of scRNA-seq data from hPSC-derived FACS-sorted D3.5 CXCR4+/GARP−/PDGFRα− PGCLCs shows that the predominant cluster express (comprising 97.2% of sorted cells) PGC markers (*NANOS3* and *TFAP2C*). **g** Bulk RNA-seq of D3.5 FACS-sorted PGCLCs (generated with either XAV939 or base media) or D3.5 CHIR99021-treated populations (lacking PGCLCs). The Pearson correlation between these samples was calculated using median expression values of all expressed genes within all three biological replicates within each condition. Statistical test: Pearson correlation with 95% confidence interval.

Other cells instead lost NANOG expression, becoming non-PGCLCs (Fig. 8d, Movie S1).

Therefore, as NANOG+ pluripotent cells differentiate into NANOG+ posterior epiblast, differentiating cells that "inherit" pluripotency factor expression from the posterior epiblast may progress forth to the germline through, at least in part, the inhibition of WNT signaling. Continued NANOG expression may thus serve as a bridge to link the pluripotent and PGC states. This mirrors staining analyses of cynomolgus macaque embryos[25], and remains to be substantiated in other species.

Finally, we sought to understand if the continuous expression of NANOG was functionally important to generate human PGCLCs. We therefore performed siRNA knockdown of *NANOG* in *NANOG-2A-YFP* reporter hESCs at different stages of differentiation (Fig. 8e) and verified *NANOG* knockdown by qPCR (Fig. S9d). *NANOG* knockdown at either D0 or D0.5 markedly decreased PGCLC formation by ~3–5-fold (Fig. 8e), reaffirming the importance of *NANOG* expression in posterior epiblast intermediates for subsequent PGCLC formation. Taken together, this suggests that the continuous expression of NANOG in cells transitioning from pluripotent state to PGCLCs is functionally important.

### Single-cell RNA-seq analysis shows that in vitro-derived PGCLCs have transcriptional similarities with in vivo-derived human fetal PGCs

Finally, we used scRNA-seq to determine whether hPSC-derived PGCLCs resemble bona fide PGCs within the human fetus. Past work affirmed similarities between human PGCLCs and PGCs using bulk-population RNA-seq[7] but did not acquire single-cell resolution. Another study used scRNA-seq to compare human PGCLCs and cynomolgus macaque PGCs[16], but did not compare them against human fetal PGCs.

A published scRNA-seq analysis of >2000 fetal germ cells (FGCs, including a subset of PGCs) from week 5-26 human fetuses[51] laid a foundation for assessing the identity of in vitro-derived PGCLCs since it offers a comprehensive roadmap for germ cell development in vivo under physiological conditions. In that study, human FGCs were classified into four sequential subsets characterized by mitosis, retinoid signaling, meiotic prophase, and oogenesis (termed FGC1 to FGC4, respectively)[51] (Fig. S10a–c). We compared human FGCs, along with fetal gonad somatic cells[51], against our hPSC-derived D3.5 PGCLCs (Fig. 9).

Hierarchical clustering revealed that hPSC-derived PGCLCs were the most similar to FGC1, which represents early-stage PGCs (Fig. 9a–c). hPSC-derived PGCLCs and FGC1 both expressed pluripotency genes including *POU5F1* and *NANOG*, as well as PGC-specific markers such as *NANOS3, SOX17, PRDM1* and *TFAP2C* (Fig. S10b). By contrast, such pluripotency and PGCs markers were turned off in later-stage FGC2, FGC3, or FGC4 populations, consistent with exit from a PGC state in vivo (Fig. S10b). hPSC-derived PGCLCs appeared to

represent an early PGC population, as they did not express markers of differentiating germline cells (e.g., those involved in retinoid signaling, oogenesis, or meiosis), which instead were expressed in FGC2, FGC3 or FGC4 (Fig. S10b). While PGCLCs most closely resembled FGC1 (Pearson correlation of *R* = 0.82), it is not possible to access human pre-migratory PGCs at earlier developmental stages[51], and it is thus possible that PGCLCs may correspond to even earlier-stage PGCs (Fig. 9c–e). Clustering, differential gene expression analysis, and gene ontology analysis revealed that PGCLCs that did not co-cluster with the rest of the FGC1 population were higher in mitochondrial gene expression and were enriched for genes linked to protein translation and cell-cell adhesion (Fig. S10d, e). By contrast, FGC1 cells that did not co-cluster with PGCLCs were enriched for cell cycle genes (Fig. S10d).

We also transcriptionally compared our hPSC-derived PGCLCs generated in monolayer culture with WNT inhibitor with previously published PGCLCs generated in the prevailing 3D differentiation system[6,16] (Fig. S11a–g). Our analysis revealed that 20% of 3D-derived PGCLCs were highly similar to PGCLCs derived from our monolayer protocol (Pearson correlation of *r* = 0.91) (Fig. S11c, d, f). Differential gene expression analysis between our monolayer PGCLCs and previously published PGCLCs in 3D[16] showed higher expression of several PGC markers (*NANOS3, TFAP2C,* and *SOX17*), and gene ontology analysis revealed enrichment of cell adhesion, cell redox, and glycolytic processes, in monolayer PGCLCs (Fig. S11f, g and Supplementary Data 3).

Taken together, this shows transcriptome-wide similarities between hPSC-derived PGCLCs in vitro and human fetal PGCs in vivo. hPSC-derived PGCLCs apparently represent an early PGC population prior to the initiation of germline differentiation and meiosis. Finally, our monolayer PGCLCs are transcriptionally similar to PGCLCs derived by the prevailing 3D differentiation protocol.

## Discussion

Expanding upon past work that successfully generated PGCLCs from hPSCs in 3D aggregates[6–8], here we report a simplified monolayer system to produce human PGCLCs and we exploit this system to provide additional insights into PGCLC specification using single-cell RNA-seq. Stem cells negotiate a series of branching lineage decisions during differentiation[27,52]. We hypothesized that at a critical lineage bifurcation, differentiating hPSCs may inadvertently stray down a non-PGC lineage. At this bifurcation, what signals promote PGCLC specification at the expense of non-PGCs?

A principal finding of this work is that temporally dynamic activation, followed by inhibition, of WNT signaling enhanced human PGCLC specification. In the first phase of differentiation, hPSCs are briefly exposed to primitive streak-inducing signals (WNT and TGFβ) for 12 h to generate candidate "posterior epiblast" cells[8]. By scRNA-seq, this intermediate cell state expresses *OCT4* and *NANOG* together with posterior epiblast/future primitive streak markers, and thus appears analogous to mouse and pig posterior

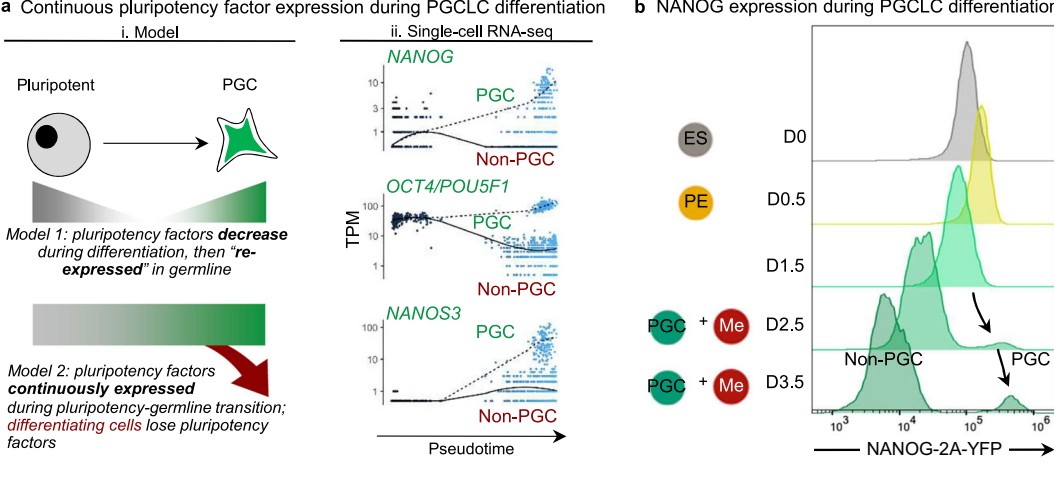

**a** Continuous pluripotency factor expression during PGCLC differentiation

i. Model

ii. Single-cell RNA-seq

Pluripotent → PGC

*Model 1: pluripotency factors **decrease** during differentiation, then "**re-expressed**" in germline*

*Model 2: pluripotency factors **continuously expressed** during pluripotency-germline transition; differentiating cells lose pluripotency factors*

**b** NANOG expression during PGCLC differentiation

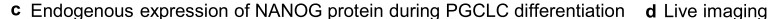

**c** Endogenous expression of NANOG protein during PGCLC differentiation

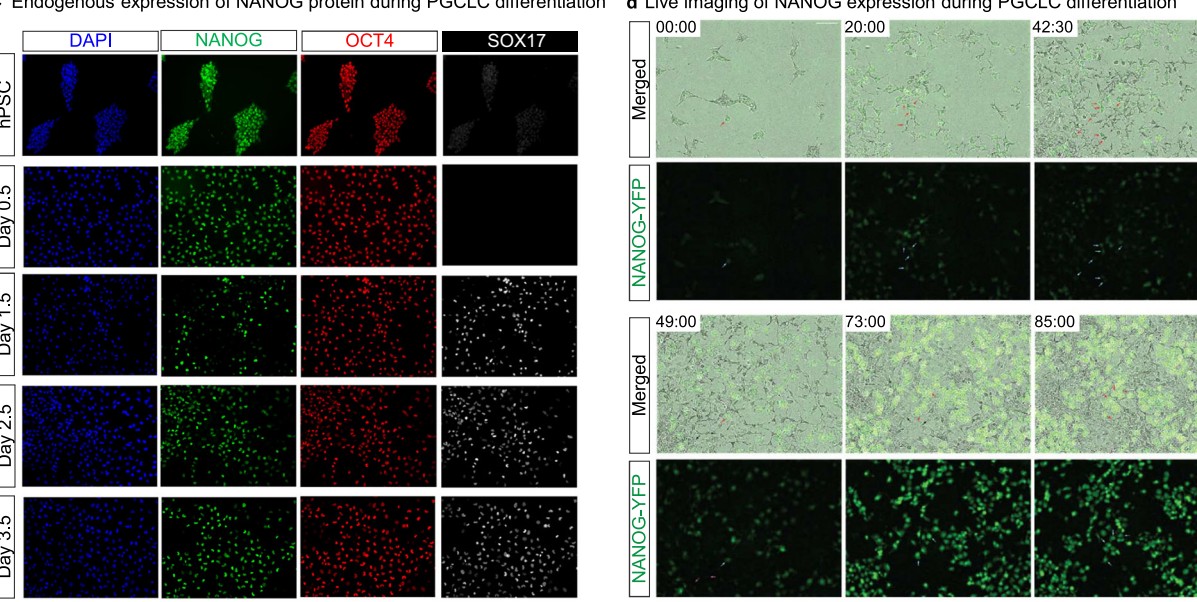

**d** Live imaging of NANOG expression during PGCLC differentiation

**e** Early NANOG expression is essential for PGCLCs generation

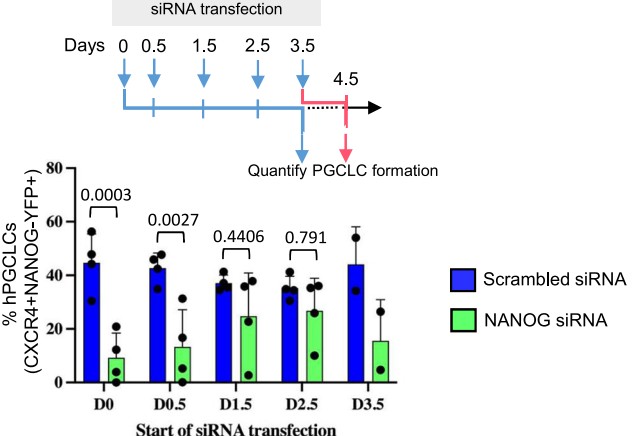

epiblast cells[8,22]. In the second phase of differentiation, these posterior epiblast cells apparently face a branching lineage choice to differentiate into PGCLCs or primitive streak. At this lineage choice, continued WNT activation specified primitive streak[8,28,29], and therefore WNT *inhibition* was critical to suppress primitive streak formation and to differentiate posterior epiblast into PGCLCs. In

this model, the PGC and somatic (primitive streak) lineages are related−yet distinct−cell-types that both arise from posterior epiblast intermediates and are segregated by mutually-exclusive signals (e.g., WNT). This temporally dynamic role for WNT signaling agrees with analyses of pig embryos[8], but further in vivo analyses are warranted to confirm various aspects of this model.

**Fig. 8 | Pluripotency factor NANOG is continuously expressed throughout hPSC-to-PGCLC differentiation. a** (i) Current models for pluripotency factor expression; (ii) Pseudotemporal analysis of single-cell RNA-seq trajectories indicates continuous expression of *NANOG* and *OCT4*. **b** Flow cytometry analysis of H9 *NANOG-YFP* hESCs shows homogeneous YFP expression at D0, D0.5, and D1.5, with a separate YFP-high PGCLC population distinguishable at D2.5 and D3.5. *n* = 2 biological replicates. Source data are provided as a Source Data file. **c** Immunostaining of endogenous NANOG, OCT4, and SOX17 protein expression in H9 hESCs from D0 to D3.5 of PGCLC differentiation. Scale bar = 100 μm. Representative images from

three independent experiments. **d** Live imaging analysis of H9 *NANOG-YFP* hESCs differentiating to PGCLCs at the indicated timepoints Scale bar = 100 μm. Representative images from three independent experiments. **e** Quantification of PGCLC differentiation efficiencies after *NANOG* siRNA knockdown at different timepoints during differentiation. Data are presented as mean values ± SEM. *n* = 4 biological replicates/group for all timepoints except for d3.5 where *n* = 2 biological replicates. Statistical test: two-way ANOVA with Šídák multiple test correction. Adjusted *P* values are shown above error bars. Source data are provided as a Source Data file.

At the transition from pluripotent to germline states, we find that pluripotency factor NANOG is continuously expressed in the transition from hPSCs to posterior epiblast to PGCLCs in response to WNT modulation. This mirrors preliminary evidence drawn from cynomolgus macaque embryos[25]. It was initially hypothesized that differentiating cells lose pluripotency factor expression, but cells allocated to the germline "re-express" such factors, thus "regaining" features of pluripotency[23,26]. By contrast, we propose that the continued expression of pluripotency factors such as NANOG may serve as a direct molecular bridge between the pluripotent and germline states; cells that lose such expression may instead differentiate into somatic cells. Our demonstration of a functional requirement of NANOG throughout the differentiation process altogether is consistent with a critical role for NANOG in mouse PGC specification in vivo and in vitro[53,54].

However, in the monolayer system, typically on average 46.3% of cells differentiate into PGCLCs in 3.5 days, across all hPSC lines. To overcome this limitation, we discovered an alternative cell-surface marker signature for PGCLCs (CXCR4⁺ PDGFRα⁻ GARP⁻) that enables their purification from multiple hPSC lines. Past markers of PGCLCs (e.g., alkaline phosphatase activity, EpCAM, ITGA6, and PDPN) were also expressed on undifferentiated hPSCs[6,7,9]. However, we find that the CXCR4⁺ PDGFRα⁻ GARP⁻ signature separates PGCLCs from undifferentiated hPSCs as well as the non-PGCLCs inadvertently generated during differentiation. Our single-cell RNA-seq analysis of FACS-sorted CXCR4⁺ PDGFRα⁻ GARP⁻ PGCLCs also reaffirmed that they are a nearly homogeneous population, and thus we foresee this surface marker sorting strategy will have broad applications in isolating PGCLCs for molecular and functional experiments. Single-cell RNA-seq analysis also revealed strong transcriptome-wide similarities between hPSC-derived CXCR4⁺ PDGFRα⁻ GARP⁻ PGCLCs and early human fetal PGCs[51], adding to past work that demonstrated similarities by bulk-population RNA-seq[7]. This thus serves to molecularly authenticate the monolayer-generated PGCLCs; however, further functional in vivo tests await, given the ethical difficulties of transplanting human PGCLCs into animal models.

Our scRNA-seq survey revealed that the remaining "mis-differentiated" non-PGCLCs are mesoderm-like cells. In both monolayer conditions (this study) as well as within 3D aggregates[6–8], our inability to generate pure PGCLCs in vitro indicates that we have an incomplete understanding of the inductive and repressive extracellular signals leading to human PGCLCs specification. Recent reports suggest that repression of *Otx2* (in mouse)[55], or overexpression of *SOX17* and *BLIMP1* (in human)[7,8], suffices to generate PGCLCs in vitro even in the absence of any exogenous signals. Delineating the upstream extracellular signals that repress *OTX2*, or that induce *SOX17* and *BLIMP1*, is therefore paramount to further enhance the efficiency of human PGC formation in vitro. Collectively, our data, together with another recent study[46], demonstrate that it is possible to generate human PGCLCs in monolayer cultures. This platform will simplify efforts to dissect the molecular mechanisms that regulate human PGCs induction and maturation and may accelerate the identification of culture conditions that will be conducive to the formation of fully functional, meiosis-competent human germ cells.

## Methods

### Human pluripotent stem cell (hPSC) culture

H1 hESCs[56], H9 hESCs[56], *NANOS3-mCherry* WIS1 hESCs[7], *SOX17-GFP* H9 hESCs[36], *NANOG-2A-YFP* H9 hESCs[50], BJC1 hiPSCs[57], BJC3 hiPSCs[57], and BIRc3 hiPSCs were routinely propagated feeder-free in mTeSR1 medium + 1% penicillin/streptomycin (StemCell Technologies) on cell culture plastics coated with Matrigel (Corning). Undifferentiated hPSCs were maintained at high quality with particular care to avoid any spontaneous differentiation, which would confound downstream differentiation.

In the *NANOS3-mCherry* hESC line, a *2A-mCherry* fluorescent reporter was inserted immediately downstream of the *NANOS3* gene without disrupting its coding sequence[7]. In the *SOX17-GFP* hESC line, a *GFP* fluorescent reporter was inserted immediately after the *SOX17* start codon, thus functionally invalidating one *SOX17* allele[36]. In the *NANOG-2A-YFP* hESC line, Cas9 RNP/AAV6-based genome editing[49] was used to insert a *2A-iCaspase9-2A-YFP* fluorescent reporter immediately downstream of the *NANOG* gene without disrupting the *NANOG* coding sequence[50].

### hPSC differentiation into PGCLCs

Undifferentiated hPSCs were maintained in mTeSR1 + 1% penicillin/streptomycin and enzymatically passaged (Accutase, 1:8–1:12 split) for differentiation. After overnight recovery in mTeSR1 + Thiazovivin (ROCK inhibitor, 5 μM), the following morning, hPSCs were briefly washed (DMEM/F12) and differentiated into posterior epiblast for 12 h (100 ng/mL Activin + 3 μM CHIR99021 + 10 μM Y-27632) in aRB27 basal media, which comprised Advanced RPMI 1640 medium supplemented with 1% B27 supplement, 0.1 mM non-essential amino acids (NEAA), 100 U/mL Penicillin + 0.1 mg/mL Streptomycin, and 2 mM L-Glutamine[8]. Subsequently, cells were washed once more (DMEM/F12) and treated with 40 ng/mL BMP4, 1 μM XAV939, and 10 μM Y-27632 for 24 h, then 100 ng/mL SCF, 50 ng/mL EGF, 1 μM XAV939, and 10 μM Y-27632 for an additional 24 hrs, and finally 40 ng/mL BMP4, 100 ng/mL SCF, 50 ng/mL EGF, 1 μM XAV939, and 10 μM Y-27632 for 24 h (all in aRB27 basal media). In certain optimization experiments, LIF and TC-S 7001 were tested, but were found to be dispensable. For comparison, published PGCLC differentiation protocols[6,8] were performed as described previously. The list of activators and inhibitors of signaling pathways used for differentiation are listed in Supplementary Data 6.

### Immunostaining

hPSCs were differentiated into PGCLCs as described above, except they were grown on Matrigel-coated glass coverslips (Fisher). Cells were washed with PBS and then fixed with 4% PFA (paraformaldehyde) for 15 min. Coverslips were then washed with PBS, permeabilized with 0.2% Triton X-100 and blocked with PBS-BT (3% BSA + 0.1% Triton X-100 + 0.02% sodium azide in PBS) for at least 30 min. Coverslips were incubated with primary antibodies diluted in PBS-BT overnight, and then washed with PBS-BT, subsequently incubated with secondary antibodies diluted in PBS-BT for 45 min, and then washed again. Finally, samples were mounted in ProLong Diamond anti-fade mountant with DAPI (Thermo Fisher Scientific). Images were acquired on a Leica

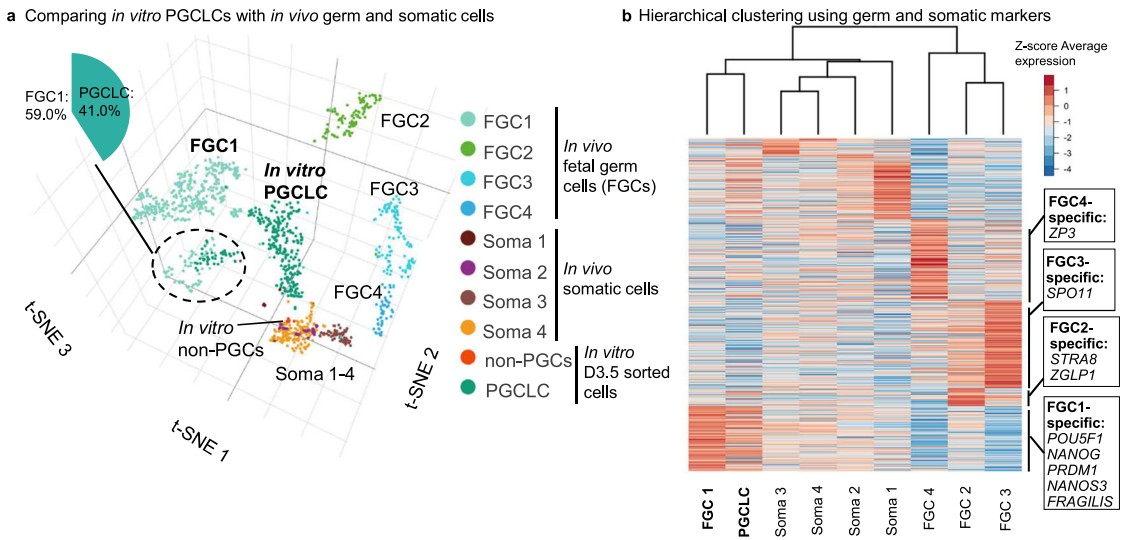

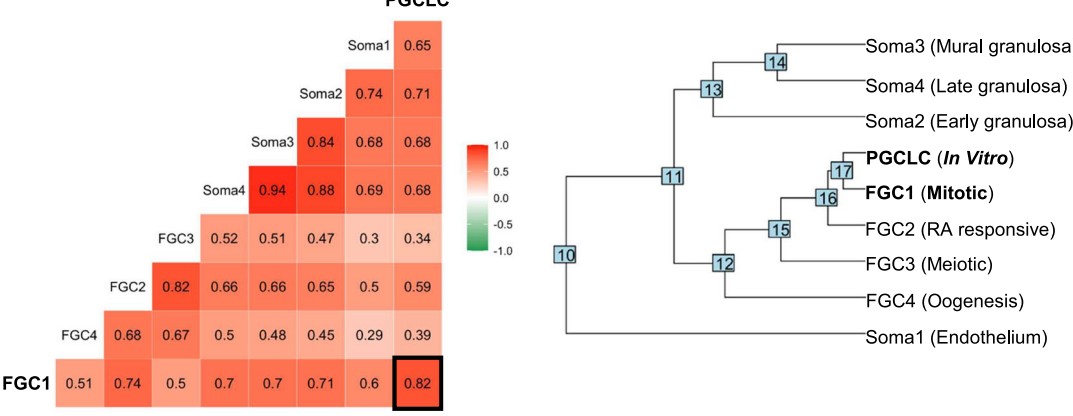

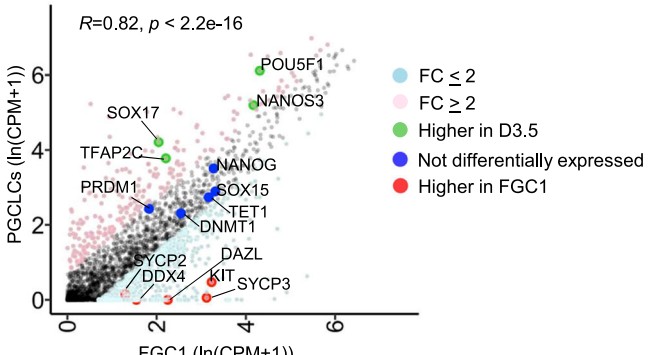

**Fig. 9 | Single-cell RNA-sequencing confirms that in vitro-derived PGCLCs resemble human fetal PGCs in vivo at the transcriptome-wide level. a** 3D integrated clustering of human female fetal germ cells clusters (FGC1-4) and somatic cells clusters (Soma1-4) with in vitro Day 3.5 sorted PGCLCs and non-PGCs with tSNE1, tSNE2, and tSNE3 dimensions. Of note, the FGC1 group clusters together with PGCLCs (dotted circle). **b** Supervised hierarchical clustering of in vivo human female fetal germ cells clusters (FGC1-4) and in vitro PGCLCs using germ cells and somatic cell-specific genes (2543 genes). **c** Pearson correlation analysis of in vivo human female fetal germ cells clusters (FGC1-4), in vivo somatic cells clusters, and Day 3.5 sorted in vitro PGCLCs. **d** Hierarchical clustering of in vivo human female fetal germ cells clusters (FGC1-4), in vivo somatic cells clusters, and in vitro PGCLCs, using all variable genes as input for the analysis. **e** Gene expression comparison between in vivo female fetal FGC1 and in vitro PGCLCs. Statistical test–Pearson correlation, 95% confidence interval (two-tailed), $p < 2.2e-16$.

---

DM4000 B (Leica Microsystems, Inc., IL, USA) equipped with a QImaging Retiga-2000R (Teledyne Photometrics, AZ, USA) digital camera using a ×40 objective, and processed using FIJI (v.1.52p)[58]. The list of antibodies used for immunostaining are listed in Supplementary Data 6.

For quantification, sorted cells were manually counted, whereas unsorted cells at D3.5 were analyzed using Biodock Online AI Nuclear Segmentation tool with analysis dashboard (www.biodock.ai).

Fluorescence intensity thresholds were empirically determined first for BLIMP1 and SOX17, and then for BLIMP1 and NANOG. Data were then analyzed on GraphPad Prism, version 9.1.

**Live cell imaging**

hPSCs were seeded, and then were transferred into an Essen IncuCyte Zoom live cell imaging station (Essen BioScience) housed within a

standard tissue culture incubator. hPSCs were then differentiated into PGCLCs as described above. Images were taken every 30 min for the entire duration of the experiment (88 h). Raw data were background corrected using the IncuCyte ZOOM Live-Cell Analysis System software and then analyzed with Fiji (v.1.52p)[58].

## High-throughput cell-surface marker screening

hPSCs or differentiated PGCLCs were dissociated (using Accutase) and plated into individual wells of four 96-well plates, each well containing a distinct antibody against a human cell surface antigen, altogether totaling 371 unique cell-surface markers across multiple 96-well plates (LEGENDScreen PE-Conjugated Human Antibody Plates; Biolegend, 700007)[28]. For each LEGENDScreen experiment, approximately 10-70 million cells of each lineage were used. High-throughput cell-surface marker staining was largely done as per the manufacturer's recommendations, and cells were stained with a viability dye (DAPI, 1.1 µM; Biolegend) prior to analysis on a CytoFLEX Flow Cytometer (Stanford Stem Cell Institute FACS Core). Stained cells were not fixed prior to FACS analysis. Sometimes, after lysophilized antibodies were reconstituted in LEGENDScreen plates they were aliquoted into a separate plate to generate replicates of antibody arrays. Undifferentiated H9 hESCs and *SOX17-GFP* H9 hESCs were used for LEGENDScreen analyses. Cell surface marker screening data is provided in Supplementary Data 1.

## Flow cytometry and fluorescence-activated cell sorting (FACS) for cell-surface marker expression

hPSCs or their differentiated derivatives were dissociated using TrypLE Express (Gibco), were washed off the plate with FACS buffer (PBS + 0.1% BSA fraction V [Gibco] + 1 mM EDTA [Gibco] + 1% penicillin/ streptomycin [Gibco]) and were pelleted by centrifugation (5 mins, 4 °C). Subsequently, cell pellets were directly resuspended in FACS buffer containing pre-diluted primary antibodies (listed below), thoroughly triturated to ensure a single-cell suspension, and primary antibody staining was conducted for 30 mins on ice. Afterwards, cells were washed with an excess of FACS buffer and pelleted again, and this was conducted one more time. Finally, washed cell pellets were resuspended in FACS buffer containing 1.1 µM DAPI (Biolegend), and were strained through a 35 µm filter. Flow cytometry and sorting were conducted on a BD FACSAria II (Stanford Stem Cell Institute FACS Core). The list of antibodies used for flow cytometry is listed in Supplementary Data 6.

## RNA extraction and reverse transcription

In general, RNA was extracted from undifferentiated or differentiated hPSC populations plated in 12-well format by lysing them with 350 µL RLT Plus Buffer per well. RNA was extracted with the RNeasy Plus Mini Kit (Qiagen) as per the manufacturer's instructions. Generally, 50–200 ng of total RNA was reverse-transcribed with the High Capacity cDNA Reverse Transcription Kit (Applied Biosystems) to generate cDNA libraries for qPCR.

## Quantitative PCR (qPCR)

Total cDNA was diluted 1:10-1:30 in $H_2O$ and qPCR was performed with the SensiFAST SYBR Hi-ROX Kit (Bioline) with 10 µL qPCR reactions per well in a 384-well plate: each individual reaction contained 5 µL 2x SensiFAST SYBR qPCR Master Mix + 4.2 µL cDNA (totaling ~120 ng of cDNA) + 0.8 µL of 10 µM primer stock (5 µM forward + 5 µM reverse primers). In general, gene-specific primer pairs for qPCR were tested for (1) specificity of amplicon amplification (only one peak on a dissociation curve) and (2) linearity of amplicon amplification (linear detection of gene expression in cDNA samples serially diluted seven times over two orders of magnitude, with 90-110% efficiency of amplification deemed acceptably linear). After qPCR plates were prepared by arraying sample-specific cDNAs and gene-specific primers

(listed below), they were sealed and briefly centrifuged (5 mins). 384-well qPCR plates and their adhesive sealing sheets were obtained from Thermo (AB1384 and AB0558, respectively). qPCR plates were run on a 7900HT Fast Real-Time PCR System (Applied Biosystems) with the following cycling parameters: initial dissociation (95 °C, 2 mins) followed by 40 cycles of amplification and SYBR signal detection (95 °C dissociation, 5 seconds; 60 °C annealing, 10 seconds; followed by 72 °C extension, 30 seconds), with a final series of steps to generate a dissociation curve at the end of each qPCR run. During qPCR data analysis, the fluorescence threshold to determine Ct values was set at the linear phase of amplification. The list of primers used for qPCR analysis, are provided in Supplementary Data 6.

## Single-cell RNA-sequencing

We performed single-cell RNA-sequencing of undifferentiated H9 hPSCs as well as differentiated D0.5 posterior epiblast, D3.5 PGCLCs (bulk population), D3.5 FACS-sorted CXCR4+ PDGFRA⁻ GARP⁻ PGCLCs, and D2 definitive endoderm populations. Cells of various stages were dissociated and washed twice in wash buffer (0.04% Bovine Serum Albumin in $Ca^{2+}/Mg^{2+}$-free PBS) and counted on the Countess II automated cell counter (Thermo Fisher). For each cell population, 10,400 cells were loaded per lane on the 10x Genomics Chromium platform[32], with the goal of capturing 6,000 cells. Cells were then processed for cDNA synthesis and library preparation using 10X Genomics Chromium Version 2 chemistry (catalog number 120234) as per the manufacturer's protocol. cDNA libraries were checked for quality on the Agilent 4200 Tape Station platform and their concentration was quantified by KAPA qPCR. Libraries were sequenced using a HiSeq 4000 (Illumina) to a depth of, at a minimum, 70,000 reads per cell.

## Single-cell RNA-seq computational analysis of hPSC-derived cell-types

Illumina base call files were converted to FASTQ files using the Cell Ranger v2.0 program. FASTQ files were then aligned to the hg19 human reference genome using Cell Ranger. The Seurat R package (v2.3.1)[59] was used for subsequent analyses. Cells from all the various timepoints were first combined into a single Seurat object. For quality control, we first filtered out low-quality cells that expressed fewer than 2500 genes; we also excluded cells that expressed more than 7,500 genes (which would imply doublets) or that expressed more than 0.15% mitochondrial genes (indicative of dead cells in this dataset). Counts were normalized and scaled by a factor of 10,000. To adjust for cell cycle effects, S phase and G2M genes were regressed out before Principal Component Analysis (PCA) was performed using the highly variable genes.

For further analyses, 1000 cells were randomly sampled from each of the 5 data sets (D0, D0.5, D3.5 sorted, D3.5 unsorted, and D2 definitive endoderm) and then combined into a new file for further analysis. The top six principal components were used for clustering using the Shared Nearest Neighbor (SNN) algorithm, which was implemented via the FindCluster function in Seurat[59]. Clusters were visualized in t-SNE dimensional reduction plots with 3-dimensional embedding. Differentially expressed genes between clusters were identified using the Wilcoxon rank sum test, which was performed via the Seurat package[59]. For all other independent library analyses, the following numbers of top principal components were used: Day 3.5 sorted library (15 principal components); Day 3.5 unsorted library (10 principal components); Day 0 vs. Day 0.5 analysis (10 principal components); Day 0.5 library (20 principal components).

Starting from genes that were differentially expressed between each cell-type, we specifically discovered transcription factors (TFs)[60], cell-surface proteins[61], and signaling ligands and receptors[62] whose expression was enriched in each cell population, using published and curated lists for each set of genes.

All single-cell RNA-seq plots were generated using Seurat[59] and ggplot2 (https://ggplot2.tidyverse.org/) R packages.

Finally, we computationally inferred a trajectory for progression from D0 hPSCs to D0.5 posterior epiblast to the D3.5 PGCLC-containing population. For this analysis, we used both unsorted and sorted D3.5 populations; the unsorted D3.5 populations contains both PGCLCs and non-PGCLCs (i.e., mesoderm-like cells), allowing us to capture the divergence between these two mutually-exclusive lineages. For trajectory inference, we only used genes expressed in at least 10 cells. For pseudotemporal ordering, Monocle version 3.0 (https://cole-trapnell-lab.github.io/monocle3/) was used, with Uniform Manifold Approximation and Projection (UMAP) reduction[63] (numbers of dimensions chosen: $n = 9$ for all cells, $n = 9$ for sub-trajectory to PGCLC, and $n = 11$ for sub-trajectory to non-PGCLCs) and PCA preprocessing implemented. For sub-trajectory analysis leading to PGCLCs, starting principal node 45 and ending principal nodes (132, 215, 220, 206, 225, 233, 221, 243 and 247) were used. For sub-trajectory leading to non-PGCLCs, starting principal node 45 and ending principal nodes (128, 127, 130, 140, 145, 148, 150, 158, 169, 168, 170, 177) were used. To identify gene modules significantly changing with pseudotime, a stringent cutoff of q value of 1e-5 and p value < 0.005 was used. List of gene modules and corresponding genes changing with pseudotime for the sub-trajectories are provided in Supplementary Data 2.

## Single-cell RNA-seq computational analysis of human fetal germ cells

Single-cell RNA-seq data of 2629 human fetal gonadal cells in vivo—including both fetal germ cells (FGCs) and gonadal somatic cells—were previously published[51] and were downloaded from Gene Expression Omnibus (GSE86146). Using the Seurat v3 platform[64], we performed quality control preprocessing on this in vivo dataset to 1) filter out genes that were expressed in fewer than 10 cells, 2) exclude low-quality cells that expressed fewer than 2000 genes, and 3) exclude low-quality cells with mitochondrial gene content greater than 5%. After these quality control steps, we obtained 2321 high-quality cells that we used for the following analysis.

Since our goal was to compare in vivo FGCs against in vitro hPSC-derived PGCLCs (and we used the female H9 hPSC line for our single-cell RNA-seq studies), we then selected female FGCs and somatic cells from the in vivo dataset[51]. A total of 992 female in vivo cells were available and used; 711 were FGCs with ages spanning from 4 weeks to 24 weeks of fetal life, while the remaining 281 were gonadal somatic cells. The study that originally reported this in vivo FGC dataset classified these FGCs into four transcriptional subclusters (FGC1, FGC2, FGC3, and FGC4)[51], and in our study, we used the same subclusters for transcriptional comparisons against hPSC-derived PGCLCs.

To compare FGCs against in vitro hPSC-derived PGCLCs, we applied the same quality control preprocessing on the FACS-sorted CXCR4+ PDGFRA− GARP− PGCLC single-cell RNA-seq dataset (with 5447 high-quality cells obtained from the original dataset of 5467 cells). We randomly down-sampled 300 of these hPSC-derived FACS-sorted PGCLCs for downstream analysis and showed that such random down-sampling of the PGCLC population did not substantially affect data quality (Fig. S10a). We then integrated the in vivo FGC and in vitro hPSC-derived PGCLC single-cell RNA-seq datasets using SCTransform[65].

We obtained the list of differentially expressed genes between the FGC1, FGC2, FGC3, and FGC4 populations in vivo from the original publication[51], with the exception that we applied a slightly more stringent statistical threshold to identify differentially expressed genes (power >0.5). For these genes that were differentially expressed between (FGC1, FGC2, FGC3, and FGC4) and somatic cells (Soma 1, Soma 2, Soma 3, and Soma 4), we also quantified the expression levels of such genes in the hPSC-derived PGCLC population (with non-PGC cells removed). This combined expression table of in vivo vs. in vitro

cells are provided in Supplementary Data 3, and expression values are presented as transcripts per million (CPM + 1), and $\log_e$ transformation was applied. To determine the similarity in gene expression profiles between hPSC-derived PGCLCs in vitro and FGC1, FGC2, FGC3, and FGC4 in vivo, we performed hierarchical clustering using the Average Linkage method and using a Euclidean distance metric, which showed that FGC1 and hPSC-derived PGCLCs clustered together. Pairwise Pearson's correlation coefficients were calculated for hPSC-derived PGCLCs in vitro, FGC1, FGC2, FGC3, FGC4, Soma 1, Soma 2, Soma 3, and Soma 4 in vivo using the GGally (1.5.0) R package on average expression derived from the AverageExpression function (Seurat package).

For comparison with another hPSC-derived PGCLC single-cell RNA-seq dataset[16], "Day 4 UCLA2" data (UCLA2 was reported to have higher efficiency than UCLA1) was downloaded from GEO database (GSE140021: GSM4202944, GSM4202950). For comparable analysis, 1000 cells were randomly sampled from each of the 5 data sets (D0, D0.5, D3.5 sorted, D3.5 unsorted and D2 definitive endoderm; total of 5000 cells) from this study and 2500 cells were randomly sampled from GSM4202944 and 2500 cells from GSM4202950 (total of 5000 cells from Day 4 UCLA2), and then integrated using Seurat V3 for further analysis (top 20 principal components were used for clustering). Differentially expressed genes between clusters are provided in Supplementary Data 4.

## siRNA knockdown

*NANOG-2A-YFP* hPSCs were transfected using ON-TARGETplus siRNA (Dharmacon) and RNAiMAX (Invitrogen) in 6-well plates, according to the manufacturer's instructions. In brief, 3 µL RNAiMAX transfection reagent and 10 pmol siRNA were separately diluted in 50 µL of Opti-MEM, and subsequently mixed and incubated for 5 min before drop-wise addition to cells in 2 mL of the desired medium. Transfections were performed correspondingly with each media change in the described differentiation protocol: at day 0, 0.5, 1.5, 2.5, and 3.5. Samples were collected at each timepoint to assess the efficiency of knockdown by qPCR (as described above). For all siRNA transfection timepoints, the purity of PGCLCs was measured by flow cytometry at day 3.5 (the differentiation endpoint). Flow cytometry measurements were carried out by both (1) measuring YFP expression as a proxy for NANOG expression and (2) staining for CXCR4, which is specific to PGCLCs. $N = 3$ biological replicates were used per group. Mean ± SEM was calculated in GraphPad Prism7 and statistical significance was calculated using Two-Way ANOVA with Šídák multiple test correction.

## Bulk RNA-seq

*NANOG-2A-YFP* hPSCs were initially differentiated to posterior epiblast in aRB27 basal medium containing 100 ng/mL Activin, 3 µM CHIR99021, and 10 µM Y-27632 for 12 h, as described above. Subsequently, these D0.5 posterior epiblast cells were further differentiated in three different types of media: "base medium", "+XAV939" (WNT inhibitor), and "+CHIR99021" (WNT agonist).

"+XAV939" media was the standard PGCLC differentiation media, as described above. The "+CHIR99021" media was as described above, with the exception that XAV939 was substituted with 3 µM CHIR99021 at each timepoint (aRB27 containing 40 ng/mL BMP4, 10 µM Y-27632, and 3 µM CHIR99021 for 24 h, then 100 ng/mL SCF, 50 ng/mL EGF, 10 µM Y-27632, and 3 µM CHIR99021 for an additional 24 hrs, and finally 40 ng/mL BMP4, 100 ng/mL SCF, 50 ng/mL EGF, 10 µM Y-27632, and 3 µM CHIR99021 for the last 24 h). Finally, "base medium" refers to the media described above, but with the omission of CHIR99021 and XAV939. To purify PGCLCs, FACS was performed to purify CXCR4+ cells at D2.5 and D3.5 of differentiation. Cells were then resuspended in Trizol reagent (Thermo Fisher Scientific) and RNA was extracted using the Quick-RNA Microprep kit (Zymo Research). 1 mg of RNA per sample ($n = 3$ biological replicates) was finally submitted for analysis to Azenta Life Sciences. Libraries were prepared by Azenta Life Sciences

and sequenced on Illumina HiSeq4000 in a 2x150bp Paired-End configuration. All raw and processed data are available through GEO database GSE210711.

## Bulk RNA-seq computational analysis of time course WNT manipulation

The raw Fastq files for each library were adaptor-trimmed with skewer (v0.2.2) and mapped to human genome GRCh38. Both coding and non-coding RNA sequences from Ensemble (release 95) were used as references. Read counts were generated using Salmon v1.7.0. Differential analysis was further performed using the DESeq2 (1.34.0) R package. A cutoff of 30 or more reads in at least two samples and rlog normalization was used agnostic of the sample group labels. To calculate differential gene expression, DESeq was applied, which estimates size factors, dispersions, and negative binomial GLM fitting and Wald statistics (two-tailed), with Benjamini-Hochberg multiple testing correction, and log fold change shrinkage. A cutoff of $P$ value $\leq 0.05$ and padj $<0.05$, and log$_2$ fold change $\geq$ or $\leq 1$ was used to calculate up- or downregulated genes. Pearson correlation was calculated using R stats package (4.1.2). 3D PCA plot was generated using r package plotly (4.10.0), heatmaps using pheatmap (1.0.12), and volcano plot using EnhancedVolcano (1.12.0).

## Reporting summary

Further information on research design is available in the Nature Portfolio Reporting Summary linked to this article.

## Data availability

The scRNA-seq and bulk RNA-seq sequencing data (raw and processed) generated in this study have been deposited in the GEO database under accession codes GSE157475 (scRNA-seq data) and GSE210711 (bulk RNA-seq data). The differential gene expression data generated in this study are provided in the Supplementary Information/Source Data file. Source data are provided in this paper.

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

## Acknowledgements

We thank Azim Surani and Naoko Irie (University of Cambridge) for advice and Virginia Winn (Stanford University) for access to the IncuCyte live cell imaging system. We also thank Yaqub Hanna (Weizmann Institute of Science), Seung Kim (Stanford University) as well as Renata Martin, and Matthew Porteus (Stanford University) for the provision of *NANOS3-mCherry* hESCs, *SOX17-GFP* hESCs, and *NANOG-YFP* hESCs, respectively. Finally, we thank Patricia Lovelace, Catherine Carswell-Crumpton, Laura Dunkin-Hubby, Liying Ou, and the Stanford Stem Cell Institute FACS Core for infrastructure support. This work was supported by a Stanford Women's Health and Sex Differences in Medicine (WHSDM) seed grant (V.S.), AFAR/Glenn Institute Breakthrough in Gerontology Award (V.S.), Milky Way Research Foundation (V.S.), NIH Director's Early Independence Award DP5OD024558 (K.M.L.), Stanford Beckman and Ludwig Centers (K.M.L.), Stanford-UC Berkeley Siebel Stem Cell Institute (L.T.A., K.M.L.), and the Anonymous, Fickel, Gilbert, and Weintz families (K.M.L.). J.L.F. was supported by National Defense Science and Engineering Graduate (NDSEG) and Stanford Honorary Bio-X Fellowships. K.M.L. is a Packard Foundation Fellow, Pew Scholar, Baxter Foundation Faculty Scholar, Human Frontier Science Program Young Investigator, and The Anthony DiGenova Endowed Faculty Scholar. V.S. is a Woods Family Faculty Scholarship in Pediatric Translational Medicine supported by the Stanford Maternal & Child Health Research Institute (MCHRI).

## Author contributions

V.S., K.M.L., G.K., S.V. and R.S. designed the experiments. G.K., R.S., M.A.P. performed cell culture, microscopy analysis, FACS, and gene expression analysis, with assistance from A.Ch., J.L.F., D.L.G. and L.T.A. S.V. performed single-cell RNA-seq experiments and all single-cell and bulk RNA-seq bioinformatic analyses. A.I.A. and A.Ci. assisted with RNA-seq data integration and analysis. All authors were involved in data interpretation and drafting and editing the manuscript.

## Competing interests

G.K., K.M.L. and V.S. are listed as inventors on provisional patent 63/032,382 and patent application PCT/US2021/034925. G.K., K.M.L., V.S. and all the other authors declare no additional competing interests.
