## [Peer Review File · Nature Communications]

Monolayer platform to generate and purify primordial germ-like cells in vitro provides insights into human germline specificationREVIEWER COMMENTS

Reviewer #1 (Remarks to the Author):

In this paper, Kang et al. established the monolayer culture platform to generate primordial germ cells from human pluripotent stem cells by modulating WNT signaling. Authors demonstrated the dynamics of gene expression during this process by scRNA-seq, in particular, continuous NANOG expression throughout the PGC induction, and identified the cell surface markers (CCR4+PDGFRA-GARP-) to enrich hPSC-derived PGCs. Finally, the authors showed the transcriptional similarities of their hPSC-derived PGCs with fetal PGCs. The major challenge of this paper is the lack of novelty. Human PGC-like cells have been studied intensively for the last 5 years and most of the findings authors presented here has been documented in the previous literature, which include the requirement of WNT signaling, continuous expression of pluripotency markers, isolation of PGCs by surface markers and single cell characterization of lineage trajectory. Of course, 2D monolayer culture provides some technological advances but overall, the findings presented in this paper are incremental at best. Moreover, how their hPSC-derived PGCs are related to fetal PGCs in vivo and PGC-like cells induced in vitro by other platforms are unclear. Although authors claim that hPSC-derived PGCs are transcriptionally similar to fetal PGCs, they also seem to be similar to somatic cells (Fig.6A). hPSC-derived PGCs might be relatively similar to FGC1 (compared with other FGCs) using selected markers (Fig.6B) but globally, they look quite different (Fig. 6D). Differences of developmental stages between PGCLCs and FGC1, and differences of RNA-seq platform are certainly needed to be taken into consideration and so "strong transcriptional similarities with fetal PGCs" does not have enough evidential support.

Reviewer #2 (Remarks to the Author):

This manuscript from Kang and colleagues reports development of a novel approach to in vitro differentiate human primordial germ cell like cells (PGCLCs) from pluripotent stem cells that obviates tedious and time-consuming 3D aggregation and performs at high efficiency. Such a "monolayer" platform would provide an extraordinary advance to the field of human germ cell specification (and any dependent downstream fields, such as in vitro gametogenesis) by greatly accelerating and simplifying the process of PGCLC derivation.

The manuscript begins with an optimization of sorts using a NANOS3-mCherry reporter hESC line. Maximal reporter expression is observed with 12hr of posteriorization signals (WNT/TGFb) and 3 days of WNT inhibition (plus some other factors). The authors concluded there was a temporally-dynamic regulation of WNT signaling required for maximal PGCLC induction. Validation was performed using a SOX17-GFP H9 hESCs, but the optimization was not repeated, so the temporal precision of the conclusion is exclusively based on one line. QPCR and marker analyses (not quantified) support the identity of the resulting cells as PGCLC, but the induction efficiency maxed out at ~1/3. Given the short time to induction, the authors used a cell surface marker screen to identify CXCR4+ and GARP- as a novel method to enrich PGCLCs without retaining PSCs or mesodermal contaminants. This was an especially strong part of the paper and demonstrated superiority to some of the prior markers employed for this purpose (CD49f, TNAP, CD326), but surprisingly, some key markers like KIT (CD117) and CD38 were essentially not detected without explanation. The induction procedure and marker selection appeared to be effective across multiple independent PSC lines, including XX and XY, hESCs and hiPSCs. To this point in the manuscript, though, the depth of molecular analysis confirming PGCLC identity was low.

Single-cell RNA-seq was performed to explore the developmental trajectory of PGCLCs compared with their 12hr epiblast-like cells, hESCs, non-PGCLC (mesoderm like) and previously reported D2 DE. Here, the analysis is rather superficial and conclusions are not fully supported by the data. First, arrows superimposed over a tSNE projection suggested developmental trajectories, but these are not

data driven and should be removed. Second, no unbiased clustering was performed to measure heterogeneity (despite clear evidence of heterogeneity), and thus, no comparisons were made among heterogeneous cells. Claims of >97% PGCLC purity after sorting were baseless. PGCLC induction a novel NANOG-2A-YFP knockin line was used to make the case that NANOG expression was continuous in germ cells, which would be an important advance, but this is only inferred from daily snapshots by flow cytometry. This conclusion could be solidified using live imaging to demonstrate continuity. Pseudotime analysis of the developmental trajectory would be improved by performing RNA velocity analysis to provide a more empirical picture of the developmental relationship between PGCLCs and precursors.

An obvious question was whether the PGCLCs produced using this novel method were similar to normal human fetal germ cells (FGCs). Comparison to existing FGC data demonstrated that the PGCLCs were unlike any of the in vivo counterparts (despite batch correction), but were most similar to the earliest population. The conclusion in the abstract of "strong transcriptional similarity" is not supported. No reasonable explanation is provided for this discrepancy, which is a concern because it raises the possibility that the differentiation is flawed. Surprisingly, no effort was made to also compare the current hPGCLCs to those made by any other method to test the relative similarity to FGCs. This is a major missed opportunity and greatly undermines the utility of this method. Also surprisingly, no effort was made to confirm the epigenetic status (5meC) of these cells, which is important given the dynamic nature of DNA methylation which is reprogrammed during germline development.

In general, I found the data to be of high quality and most of the conclusions were well supported by data (with the exceptions noted above). If the concerns raised can be addressed, this manuscript would make an outstanding contribution to the toolbox for the investigation of the early human germline (which is otherwise inaccessible) and for production of cells to develop in vitro gametes.

The following specific criticisms should also be addressed:

1. Title – for accuracy, the term PGCLC should be used.
2. Fig. 1A-C – very nice presentation!
3. Fig. 1D – unbiased clustering should be performed to support the text which indicates these D0.5 cells were "fairly homogeneous," but this semiquantitative term should be replaced by a quantitative readout.
4. Fig.2D – these data should be quantified.
5. Fig. 3A – where are KIT (CD117) and CD9?
6. Fig. 4C – these images are nice, but fail to adequately support the conclusion that the sorted cells ubiquitously expressed PGC markers. Quantification of many cells is needed (not just a single field) and suggest using flow cytometry.
7. Fig.5 – what is the replication of the scRNA-seq in this experiment?
8. Fig. 5A and F – no evidence provided for the arrows.
9. Fig. 6 – it is not clear why PGCLCs from an XY line was not also compared.
10. Fig. S2 –significant changes are not noted.
11. Methods – How were qPCR primers designed and validated?
12. Methods – how was the number of PCs for single-cell RNA-seq analyses determined?

Reviewer #3 (Remarks to the Author):

The authors developed a new method for PGCLC induction from human PSCs, in which the cells were always maintained in monolayer instead of the conventional 3D aggregate culture. In addition, they found that hPGCLCs were efficiently purified by flow cytometry by using a novel combination of cell-surface markers (CXCR4+; PDGFRA-; GFAP-). By using this culture method, they showed the

importance of repression of WNT signaling after its initial temporal activation for efficient PGC specification, and also found continuous expression of pluripotency transcription factors, OCT4 and NANOG during PGCLC induction from PSCs. They also claimed that the induced PGCLCs showed similar transcriptional characteristic as that of in vivo early PGCs.

This study provides some additional insights concerning hPGC specification in vitro, and monolayer culture for PGCLC induction has great merit for studying PGC specification, though induction of hPGCLCs from hPSCs in culture has already been studied well. The authors' claims are often not supported by the presented data as described below.

Specific comments:

1. A critical role of WNT signal repression for PGC specification is interesting, but the importance of WNT in the early phase of hESC differentiation to PGCLCs has been reported. Novelty of the finding in this study concerning additional functional insight of WNT signal in PGC specification is limited.
2. The authors also found that NANOG and OCT4 was continuously expressed during the course of PGCLC differentiation from hESCs (Fig. 5F, Fig. S5, page 8-9), but functional significance of their expression is not shown.
3. Previous studies showed that hESC-derived PGCLCs were efficiently purified by using combination of cell-surface markers such as EpCAM and Integrin $\alpha 6$, and merits of the presented methods in this study is unclear.
4. Although the authors mentioned in page 4 that 'However, D0.5 cells expressed posterior epiblast/primitive streak markers at lower levels compared to D1 primitive streak cells that were generated by 24 hours exposure to posteriorizing signals (Fig. 1e)', the expression of FGF8 is not lower in the D0.5 cells than that in D1 PS cells.
5. In Fig. S2B, please show the effect of the conventional concentration of BMP4 (200 ng/ml) for PGCLC induction in addition to that of the lower concentration (20 ng/ml).
6. In page 5, the authors mentioned that 'PGCLCs generated using our protocol did not express markers of endoderm (FOXA2, HHEX) or extraembryonic fate (CDX2), thus reaffirming their lineage specificity (Fig. S2e)', but Fig.S2e seems to show their expression in D3.5 PGCLCs. They also said that in the same page that 'Our monolayer differentiation protocol generated SOX17-GFP+ PGCLCs that expressed NANOS3 but not endodermal marker FOXA2 (Fig. S2f,g)', but Fig.S2F again shows significant expression of FOXA2 in D3.5 PGCLCs.
7. In page 6, the authors mentioned that 'the mesodermal markers PDGFR α /CD140A (Kattman et al., 2011) and GARP/LRRC32 (Loh et al., 2016) were expressed on the D3.5 non-PGCLCs (Fig. 3a,b)', but Fig.3b shows no PDGFR α -positive cells in mCherry-negative cells at least by the presented gating.
8. In the last paragraph in page 6, he authors mentioned that 'hPSC-derived CXCR4+ PDGFR α - GARP- PGCLCs upregulated of hallmark PGC markers without substantial expression of endodermal or mesodermal 6 markers (Fig. 4b, Fig. S3b)', but the figures show substantial expression of the somatic markers in D3.5 PGCLCs compared with that in ESC and D3.5 bulk cells.
9. In Fig.6, comparison of transcriptome of D3.5 PGCLCs in the monolayer culture with that of PGCLCs in 3D aggregates should be informative to discuss their possible differences.
10. The authors claimed that 'Single-cell RNA-seq analysis also revealed strong transcriptome wide similarities between hPSC-derived CXCR4+ PDGFR α -GARP- PGCLCs and early human fetal PGCs (Li et al., 2017)' in page 11, but the transcriptome in Fig. 6 shows distinct features of D3.5 PGCLCs and of FGCs.

RESPONSE TO REVIEWER COMMENTS

We thank the reviewers for their valuable and constructive comments. We have conducted extensive additional experiments and bioinformatics analysis that confirm, validate, and further enhance our original observations. We hope the reviewers will find the revised manuscript acceptable for publication.

Below we provide the original reviewers' comments (in black) and our responses (in blue).

Reviewer #1 (Remarks to the Author):

In this paper, Kang et al. established the monolayer culture platform to generate primordial germ cells from human pluripotent stem cells by modulating WNT signaling. Authors demonstrated the dynamics of gene expression during this process by scRNA-seq, in particular, continuous NANOG expression throughout the PGC induction, and identified the cell surface markers (CCR4+PDGFRA-GARP-) to enrich hPSC-derived PGCs. Finally, the authors showed the transcriptional similarities of their hPSC-derived PGCs with fetal PGCs. The major challenge of this paper is the lack of novelty. Human PGC-like cells have been studied intensively for the last 5 years and most of the findings authors presented here has been documented in the previous literature, which include the requirement of WNT signaling, continuous expression of pluripotency markers, isolation of PGCs by surface markers and single cell characterization of lineage trajectory. Of course, 2D monolayer culture provides some technological advances but overall, the findings presented in this paper are incremental at best.

Response: We thank the reviewer for the constructive comments. We respectfully disagree, at least in part, with some of the reviewer's conclusions. Below we provide our perspective and the additional experimental evidence point by point.

1. WNT signaling. The reviewer is right in stating that previous work has already shown that WNT signaling is required for PGCLC induction, but this is not the point we are making. Here we show that the **temporal dynamics** of WNT signaling are crucially important. In previous work (Sasaki et al., 2015; Kobayashi et al., 2017), WNT was activated in the first phase of differentiation for 12-42 hours, but here we show that **WNT inhibition is required in the second phase of differentiation (a new finding in our study)**. In past studies, beyond WNT activation in the first phase, it was unclear whether WNT was continually required or not at later stages. Here we show something substantially new: after the initial activation of WNT signaling for 12 hours (which generates posterior epiblast cells), the continued activation of WNT generated primitive streak cells. By contrast, WNT **inhibition** generated PGCLCs. This result was confirmed across three additional hESC lines (H1, H9, and H9-NANOG-YFP; new data available in Fig. S1).

It should also be noted that we tried two conditions: simply omitting exogenous WNT from the culture, or actively suppressing WNT signaling by small molecules. Active WNT suppression generated PGCLCs more efficiently than simply omitting WNT: this implies that some cells in the culture endogenously produce WNTs, thus reducing the efficiency of differentiation. However, given that we cannot achieve 100% efficient PGCLC induction, we conclude that WNT inhibition is necessary but not sufficient for PGCLC induction. Other signaling pathways, yet to be discovered, likely also have an important role.

2. Continuous pluripotency factor expression. It was never conclusively shown before in human that, throughout each intermediate step of differentiation from pluripotent stem cells to PGCLCs, cells continuously to express pluripotency markers. Here we conclusively demonstrate—through live-imaging of a NANOG-YFP reporter (Movie S1 and Snapshots in Fig. 6d), new immunostaining for OCT4 and NANOG (New Fig. 6c in the manuscript and here for your convenience), and single-cell RNA-seq—that pluripotency genes expression persists in the cells that progressively become PGCLCs. This result may seem “expected” to an expert, but it was never conclusively shown before in human cells. As a matter of fact, a prevailing hypothesis is that cells first differentiate to primitive streak and then re-acquire pluripotency genes expression. Here we answer this long-standing question and show that the pluripotency cells that become PGCLCs

Fig. 6 (newly incorporated in revised manuscript): Immunostaining shows continued NANOG and OCT4 expression throughout the transition from hPSCs (day 0) to the posterior epiblast (day 0.5) to PGCLCs (day 3.5).

continue to express OCT4 and NANOG. The mechanistic relevance of these findings is intriguing and important and will be explored in a follow up study.

Fig. 6c (new): Live imaging of *NANOG-2A-YFP* reporter hPSCs throughout PGCLC differentiation. 00:00 = undifferentiated hPSCs.

Moreover, how their hPSC-derived PGCs are related to fetal PGCs in vivo and PGC-like cells induced in vitro by other platforms are unclear.

Response: As the reviewer suggested, we conducted single-cell RNA-seq and found that PGCLCs generated in our simplified 2D protocol are transcriptionally similar to PGCLCs generated in a previous 3D protocol (by Amander Clark's group; Chen et al., 2019; *Cell Reports*) (Fig. S10). Our PGCLCs cluster together with Chen et al.'s PGCLCs (Fig. S10a), and show a high transcriptional correlation with them ($r = 0.91$, Fig. S10g). The major transcriptional differences in our vs. Chen et al.'s PGCLCs were mainly in mitochondrial and ribosomal genes (Table S3). Gene ontology analyses showed that "cell adhesion", "cell redox homeostasis" and "glycolytic process"-associated genes were upregulated in our PGCLCs, whereas "Translation initiation" and "mRNA splicing"-associated genes were upregulated in Chen et al. PGCLCs (Fig. S10h). Altogether, we conclude that PGCLCs generated in 2D vs. 3D conditions are transcriptionally similar, with our 2D differentiation platform offering certain practical advantages.

Fig. S10a,b (new): Single-cell RNA-seq shows that PGCLCs generated in this study transcriptionally cluster with previously-published, 3D-induced PGCLCs from Amander Clark's group (Chen et al., 2019; *Cell Reports*).

The major transcriptional differences in our vs. Chen et al.'s PGCLCs were mainly in mitochondrial and ribosomal genes (Table S3). Gene ontology analyses showed that "cell adhesion", "cell redox homeostasis" and "glycolytic process"-associated genes were upregulated in our PGCLCs, whereas "Translation initiation" and "mRNA splicing"-associated genes were upregulated in Chen et al. PGCLCs (Fig. S10h). Altogether, we conclude that PGCLCs generated in 2D vs. 3D conditions are transcriptionally similar, with our 2D differentiation platform offering certain practical advantages.

Fig. 10f-h (new): (f) PGCLCs from this study and Chen *et al.* both express *NANOS3* and *TFAP2C*. (g) Transcriptome-wide analysis of PGCLCs derived from this study or Chen *et al.* show overall high concordance ($R=0.91$). (h) Gene ontology analysis of genes upregulated or downregulated in this study's PGCLCs by comparison to Chen *et al.*'s PGCLCs.

Although authors claim that hPSC-derived PGCs are transcriptionally similar to fetal PGCs, they also seem to be similar to somatic cells (Fig.6A).

Response: As the reviewer suggested, we have now included the *in vivo* somatic cells (Li *et al.*, 2017; *Cell Stem Cell*) in our correlation analysis. Our *in vitro* PGCLCs are highly correlated with FGC1 ($R=0.82$), but less so with somatic cells ($R=0.65-0.71$) (Fig. 6e).

hPSC-derived PGCs might be relatively similar to FGC1 (compared with other FGCs) using selected markers (Fig.6B) but globally, they look quite different (Fig. 6D).

Response: To address the reviewer's concern that we only analyzed selected marker genes, we have now performed a broader analysis of all variable genes across the transcriptome (Fig. 6c,d). This hierarchical clustering still clearly shows that our *in vitro* PGCLCs cluster closest with *in vivo* FGC1 (Fig. 6c,d).

In a 3-dimensional t-SNE plot, a subset of FGC1 cells cluster with PGCLCs (Fig. 6b). In this cluster 4, 41% of the cells are PGCLCs and remaining 59% are from FGC1. (Further analysis is presented in Fig. S9.)

Differences of developmental stages between PGCLCs and FGC1, and differences of RNA-seq platform are certainly needed to be taken into consideration and so "strong transcriptional similarities with fetal PGCs" does not have enough evidential support.

Response: As the reviewer aptly pointed out, *in vitro* PGCLCs and *in vivo* FGC1 are not identical, as they likely reflect different developmental stages: most of the FGC1 analyzed by Li *et al.* have already colonized the fetal gonad, whereas *in vitro* PGCLCs likely correspond to pre-migratory PGCs that are incipiently emerging from pluripotent cells. It is also notable that fetal germ cells such as FGC1 have been induced, migrated, and colonized the gonads in a "perfect" *in vivo* environment. We now explicitly address this in the main text.

That being said, our analysis reveals that PGCLCs are most similar to FGC1, by comparison to FGC2-4 (which correspond to differentiating germ cells) or somatic cells (Fig. 6c,d). PGCLCs do not yet express more "mature" germ cell markers such as *DAZL* (Fig. 6e). The transcriptome-wide correlation of PGCLCs and FGC1 is fairly high ($R=0.82$, Fig. 7e), again reifying the similarities between these cell-types.

To our knowledge, our study is the first to compare *in vitro*-derived human PGCLCs to *in vivo*-derived human PGCs using single-cell RNA-seq. In prior studies, bulk RNA-seq comparisons were performed (e.g., Irie *et al.*, 2014; *Cell*), or alternatively, human PGCLCs were compared to *non-human primate* fetal PGCs (Chen *et al.*, 2019; *Cell Reports*). We propose human fetal PGCs (Li *et al.*, 2017; *Cell Stem Cell*) are the most appropriate comparator for human PGCLCs.

To minimize RNA-seq platform differences, we applied standard methods accepted in the field (as described in the Methods section). In brief, datasets were first normalized using the “Normalize Data” function in Seurat. Counts Per Million (CPM) values were log transformed ($\ln(\text{CPM}+1)$). They were then integrated using *sctransform* (Hafemeister and Satija, 2019), a well-established and frequently-utilized method in the Seurat V3 package to integrate data generated using different single-cell RNA-seq platforms, based on regularized negative binomial regression.

Fig. 7

A Comparing *in vitro* PGCLCs with *in vivo* germ and somatic cells

B Hierarchical clustering using germ and somatic markers

C Correlation of *in vitro* PGCLCs with *in vivo* germ and somatic cells

D Clustering of *in vitro* PGCLCs with *in vivo* germ and somatic cells

E Gene expression comparison of *in vivo* FGC1 and *in vitro* PGCLCs

Fig. 7 (largely new): (a) Single-cell RNA-seq comparison of *in vitro*-derived day 3.5 PGCLCs and non-PGCLCs with *in vivo* fetal germ and somatic cells (Li et al., 2017; *Cell Stem Cell*), displayed via 3-dimensional t-SNE. (b) Hierarchical clustering using germ and somatic markers. (c) Transcriptome-wide correlation and (d) hierarchical clustering using all variable genes. (e) Differentially expressed genes between *in vitro* PGCLCs and *in vivo* FGC1.

Reviewer #2 (Remarks to the Author):

This manuscript from Kang and colleagues reports development of a novel approach to in vitro differentiate human primordial germ cell like cells (PGCLCs) from pluripotent stem cells that obviates tedious and time-consuming 3D aggregation and performs at high efficiency. Such a “monolayer” platform would provide an extraordinary advance to the field of human germ cell specification (and any dependent downstream fields, such as in vitro gametogenesis) by greatly accelerating and simplifying the process of PGCLC derivation.

Response: We thank the reviewer for their enthusiasm. We concur that a simplified monolayer platform to generate PGCLCs would dramatically simplify the study of human gametogenesis and have broad repercussions for downstream applications.

The manuscript begins with an optimization of sorts using a NANOS3-mCherry reporter hESC line. Maximal reporter expression is observed with 12hr of posteriorization signals (WNT/TGFb) and 3 days of WNT inhibition (plus some other factors). The authors concluded there was a temporally dynamic regulation of WNT signaling required for maximal PGCLC induction. Validation was performed using a SOX17-GFP H9 hESCs, but the optimization was not repeated, so the temporal precision of the conclusion is exclusively based on one line.

Response: As the reviewer suggested, we confirmed that 12-hour WNT activation followed by WNT inhibition was consistently beneficial across 3 additional hPSC lines: H1, H9, and H9 NANOG-YFP (Fig. S1b). The experiments were repeated for three independent biological replicates for each of the 3 new lines.

QPCR and marker analyses (not quantified) support the identity of the resulting cells as PGCLC, but the induction efficiency maxed out at ~1/3.

Response: As the reviewer suggested, we have now quantified the purity of PGCLCs through triple immunostaining for BLIMP1, SOX17, and NANOG (now added to Fig. 4). This was performed across 2 independent hPSC lines (H1 and in H9). The purity of PGCLCs quantified by immunostaining (Fig. 4b) was similar to our flow cytometry-based quantification (for CXCR4+/PDGFRA-/GARP- cells, Fig. 4a).

Given the short time to induction, the authors used a cell surface marker screen to identify CXCR4+ and GARP- as a novel method to enrich PGCLCs without retaining PSCs or mesodermal contaminants. This was an especially strong demonstrated superiority to some of the prior markers (CD49f, TNAP, CD326), but surprisingly, some key markers like KIT (CD117) and CD38 were essentially not detected without explanation.

Fig. S1b (new): 3 different hPSC lines were differentiated into posterior epiblast for 6, 12 and 24 hours. Subsequently, they were differentiated towards PGCLCs for 3 additional days and analyzed by flow cytometry.

Fig. 2e (new): Quantified immunostaining of BLIMP1+ SOX17+ NANOG+ PGCLCs

Fig. 4a (new): Quantified immunostaining of FACS-CXCR4+ PDGFRA-/GARP- sorted PGCLCs.

Response: We thank the reviewer for their complimentary remarks regarding the utility of the PGCLC cell-surface markers that we identified. As the reviewer noted, we did not detect high levels of KIT or CD38 in our PGCLCs (**Fig. 3, Fig. S3**). Our rapid, 3.5-day differentiation protocol may generate PGCLCs that are at a slightly earlier developmental stage than other protocols. Indeed, preliminary experiments by a collaborator suggest that continued culture of our PGCLCs (until day 5 of hPSC differentiation) leads to CD38 upregulation. This will be further explored in subsequent studies.

The induction procedure and marker selection appeared to be effective across multiple independent PSC lines, including XX and XY, hESCs and hiPSCs. To this point in the manuscript, though, the depth of molecular analysis confirming PGCLC identity was low. Single-cell RNA-seq was performed to explore the developmental trajectory of PGCLCs compared with their 12hr epiblast-like cells, hESCs, non-PGCLC (mesoderm like) and previously reported D2 DE. Here, the analysis is rather superficial and conclusions are not fully supported by the data. First, arrows superimposed over a tSNE projection suggested developmental trajectories, but these are not data driven and should be removed.

Response: The arrows superimposed on the t-SNE projection are based on the actual time at which cells were collected during differentiation (e.g., day 0, day 0.5, day 2 or day 3.5). Pseudotemporal analysis also supports the projected arrows (**Fig. S8a,c**).

Fig. S7 (new): (a) Single-cell RNA-seq of FACS-sorted CXCR4+ PDGFRA- GARP- PGCLCs reveals 4 clusters. (b) Abundance, (c) quality control, and (d) marker expression across all 4 clusters.

Second, no unbiased clustering was performed to measure heterogeneity (despite clear evidence of heterogeneity), and thus, no comparisons were made among heterogeneous cells.

Response: As the reviewer suggested, we subclustered our FACS-sorted CXCR4+ PDGFRA- GARP- PGCLCs to discover any putative heterogeneity (**Fig. S7a,b**). 2.8% of cells were PDGFRA+ mesoderm-like cells (likely contaminating cells from our FACS sorting), and remaining 3 subsets comprised PGCLCs expressing *NANOS3*, *TFAP2C* and *KLF4* (**Fig. 5b, Fig. S4c, S7a,d**). The only features distinguishing the 3 PGCLC subtypes were cell cycle genes and higher expression of *EDN1* and *TFAP2A* in a small subset of PGCLCs (**Fig. S7d**). We conclude from these data that FACS-sorted CXCR4+ PDGFRA- GARP- PGCLCs are fairly homogenous with regard to PGC-specific markers but are heterogeneous with regard to cell-cycle status. Importantly, PGCLCs are distinct from non-PGCLCs that are mesoderm-like cells as shown by the expression of *HAND1* and other mesodermal markers. The list of top differentially-expressed genes between these cellular subtypes is in **Fig. S7e**.

Claims of >97% PGCLC purity after sorting were baseless.

Response: We used two separate means—single-cell RNA-seq and immunostaining—to rigorously interrogate the purity of FACS-sorted CXCR4+ PDGFRA- GARP- PGCLCs. *First*, single-cell RNA-seq of FACS-sorted cells showed that 97.2% of cells formed a large cluster expressing PGCLC markers (*NANOS3*, *TFAP2C*, **Fig. 5d**), with the remaining 2.8% of cells expressing mesodermal markers such as *PDGFRA* (**Fig. S6**). This may be due to incomplete (i.e., not 100%) post-sort purity after FACS sorting. *Second*, we performed immunostaining and found that >90% of FACS-sorted CXCR4+ PDGFRA- GARP- cells were BLIMP1+ SOX17+ NANOG+ triple-positive (**Fig. 4b**). Taken together, this validates that our FACS sorting strategy isolates a nearly-pure population of PGCLCs.

PGCLC induction a novel NANOG-2A-YFP knockin line was used to make the case that NANOG expression was continuous in germ cells, which would be an important advance, but this is only inferred from daily snapshots by flow cytometry. This conclusion could be solidified using live imaging to demonstrate continuity.

Response: We thank the reviewer for the excellent suggestion to perform NANOG-2A-YFP live imaging, which is now included in **Movie S1**. Snapshots in **Fig. 6c** show the continuous expression of NANOG throughout differentiation, which actually increases from undifferentiated hPSCs (00:00) to later timepoints. This is also corroborated by NANOG immunostaining (**Fig. 6d**). Our flow cytometry shows NANOG expression is homogeneous in hESCs and posterior epiblast (D0.5), but subsequently is lost in a subpopulation of cells (presumably the mesodermal contaminants), while it increases in PGCLCs (**Fig. 6b**).

Fig. 6c (new): Live imaging of NANOG-2A-YFP reporter hPSCs throughout PGCLC differentiation. 00:00 = undifferentiated hPSCs.

Pseudotime analysis of the developmental trajectory would be improved by performing RNA velocity analysis to provide a more empirical picture of the developmental relationship between PGCLCs and precursors.

Response: We thank the reviewer for this very crucial suggestion. We have conducted pseudotime analysis, which supports our inferred differentiation trajectories (**Fig. S8a,b**). We show the two different trajectories taken by the cells, diverging from Day 0.5 to PGCLCs vs. non-PGCLCs (mesoderm-like cells) (**Fig. 8b**). We now also show the gene modules that change with pseudotime on the two independent trajectories (**Table S2**).

Fig. S8a,b: (a) Pseudotemporal ordering of hPSCs differentiating through posterior epiblast intermediate into PGCLC vs. non-PGC fates. (b) Trajectory analysis of PGCLC vs. non-PGC differentiation paths.

An obvious question was whether the PGCLCs produced using this novel method were similar to normal human fetal germ cells (FGCs). Comparison to existing FGC data demonstrated that the PGCLCs were unlike any of the in vivo counterparts (despite batch correction), but were most similar to the earliest population. The conclusion in the abstract of “strong transcriptional similarity” is not supported.

Response: As the reviewer requested, we removed the phrase “strong transcriptional similarity” from the Abstract. *In vitro* PGCLCs are most highly correlated with *in vivo* FGC1 (instead of differentiating germline cells FGC2-4 or somatic cells; Fig. 7c). Despite an overall correlation of $R=0.82$ between *in vitro* PGCLCs and *in vivo* FGC1, there are still differences between these two cell-types (Fig. 7e).

No reasonable explanation is provided for this discrepancy, which is a concern because it raises the possibility that the differentiation is flawed.

Response: Thank you for raising this point. As the reviewer aptly pointed out, *in vitro* PGCLCs and *in vivo* FGC1 are not identical, as they likely reflect different developmental stages: the FGC1 analyzed by Li et al. have already colonized the fetal gonad, whereas *in vitro* PGCLCs likely correspond to pre-migratory PGCs that are incipiently emerging from pluripotent cells. PGCLCs do not yet express more “mature” germ cell markers such as *DAZL* (Fig. 6e). It is also notable that fetal germ cells such as FGC1 have been induced, migrated, and colonized the gonads in a “perfect” *in vivo* environment. We now explicitly address this in the main text.

Surprisingly, no effort was made to also compare the current hPGCLCs to those made by any other method to test the relative similarity to FGCs. This is a major missed opportunity and greatly undermines the utility of this method.

Response: We have now compared single-cell RNA-seq datasets and find that PGCLCs generated in our simplified 2D protocol are transcriptionally similar to PGCLCs generated in a previous 3D protocol (by Amander Clark’s group; Chen et al., 2019; *Cell Reports*) (Fig. S10). Our PGCLCs cluster together with Chen et al.’s PGCLCs (Fig. S10a) and show a high transcriptional correlation with them ($r = 0.91$, Fig. S10g).

Also, surprisingly, no effort was made to confirm the epigenetic status (5meC) of these cells, which is important given the dynamic nature of DNA methylation which is reprogrammed during germline development.

Fig. S2e (new): Immunostaining of hPSC-derived day 3.5 populations, showing expression of 5-hydroxymethylcytosine relative to PGC markers SOX17 and TFAP2G.

Response: To address the reviewer’s comment, we conducted immunostaining for 5-hydroxymethylcytosine, which was elevated in $SOX17^+ TFAP2G^+$ PGCLCs compared to the surrounding non-PGCLCs (Fig. S2e). Additionally, single-cell RNA-seq shows that *in vitro* PGCLCs and *in vivo* germ cells (FGC1) express similar levels of *TET1* (Fig. 7e).

In general, I found the data to be of high quality and most of the conclusions were well supported by data (with the exceptions noted above). If the concerns raised can be addressed, this manuscript would make an outstanding contribution to the toolbox for the investigation of the early human germline (which is otherwise inaccessible) and for production of cells to develop *in vitro* gametes.

Response: We thank the reviewer for their valuable comments and hope that we have addressed their requests.

The following specific criticisms should also be addressed:

1. Title – for accuracy, the term PGCLC should be used.

Response: Corrected, thank you.

2. Fig. 1A-C – very nice presentation!

Response: Thank you!

3. Fig. 1D – unbiased clustering should be performed to support the text which indicates these D0.5 cells were “fairly homogeneous,” but this semiquantitative term should be replaced by a quantitative readout.

Response: In Fig. S1e,f we performed a new single-cell RNA-seq subclustering analysis of the day 0.5 posterior epiblast population and did not find extensive heterogeneity.

4. Fig.2D – these data should be quantified.

Response: The percentage of BLIMP1+ SOX17+ NANOG+ PGCLCs generated across 2 hPSC lines (H1 and H9) is now quantified in Fig. 2e.

5. Fig. 3A – where are KIT (CD117) and CD9?

Response: We thank the reviewer for the suggestion, and have now highlighted CD9 and CD117 with orange arrows.

6. Fig. 4C – these images are nice, but fail to adequately support the conclusion that the sorted cells ubiquitously expressed PGC markers. Quantification of many cells is needed (not just a single field) and suggest using flow cytometry.

Response: Thank you for suggesting this quantification of our immunostaining images, which shows that >90% of FACS-sorted CXCR4+ PDGFRA- GARP- cells were BLIMP1+ SOX17+ NANOG+ triple-positive PGCLCs (Fig. #).

7. Fig.5 – what is the replication of the scRNA-seq in this experiment?

Response: Two separate pools of hPSCs were differentiated using our monolayer protocol and then pooled to perform single-cell RNA-seq. Due to experimental feasibility, we only performed single-cell RNA-seq on 1 independent experiment.

8. Fig. 5A and F – no evidence provided for the arrows.

Response: The arrows superimposed on the t-SNE projection are based on the actual time at which cells were collected during differentiation (e.g., day 0, day 0.5, day 2 or day 3.5). Pseudotemporal analysis also supports the projected arrows (Fig. S8a,c).

9. Fig. 6 – it is not clear why PGCLCs from an XY line was not also compared.

Response: We only compared XX cells, as we performed scRNA-seq on H9 hESC-derived cells (which are XX). To keep the analysis consistent and to avoid biological sex as a confounding factor, we only analyzed the XX cells published in Li et al.

10. Fig. S2 –significant changes are not noted.

Response: have performed statistical analysis and added the information to the Figure.

11. Methods – How were qPCR primers designed and validated?

Response: qPCR primer sequences were largely based on previous publications. Quality control criteria include a single peak in the melt curve, indicating amplification of a single DNA product.

12. Methods – how was the number of PCs for single-cell RNA-seq analyses determined?

Response: We chose the top significant PCs based on the ElbowPlot function in Seurat, which plots the standard deviations of the PC components, allowing us to identify the most meaningful set of PCs for use in downstream analyses.

Reviewer #3 (Remarks to the Author):

The authors developed a new method for PGCLC induction from human PSCs, in which the cells were always maintained in monolayer instead of the conventional 3D aggregate culture. In addition, they found that hPGCLCs were efficiently purified by flow cytometry by using a novel combination of cell-surface markers (CXCR4+; PDGFRA-; GFAP-). By using this culture method, they showed the importance of repression of WNT signaling after its initial temporal activation for efficient PGC specification, and also found continuous expression of pluripotency transcription factors, OCT4 and NANOG during PGCLC induction from PSCs. They also claimed that the induced PGCLCs showed similar transcriptional characteristic as that of in vivo early PGCs.

This study provides some additional insights concerning hPGC specification in vitro, and monolayer culture for PGCLC induction has great merit for studying PGC specification, though induction of hPGCLCs from hPSCs in culture has already been studied well. The authors' claims are often not supported by the presented data as described below.

Specific comments:

1. A critical role of WNT signal repression for PGC specification is interesting, but the importance of WNT in the early phase of hESC differentiation to PGCLCs has been reported. Novelty of the finding in this study concerning additional functional insight of WNT signal in PGC specification is limited.

Response: The reviewer is right in stating that previous work has already shown that WNT signaling is required for PGCLC induction, but this is not the point we are making. Here show that the **temporal dynamics** of WNT signaling are crucially important. In previous work (Sasaki et al., 2015; Kobayashi et al., 2017), WNT was activated in the first phase of differentiation for 12-42 hours, but here we show that **WNT inhibition is required in the second phase of differentiation (a new finding in our study)**. In past studies, beyond WNT activation in the first phase, it was unclear whether WNT was continually required or not at later stages. Here we show something substantially new: after the initial activation of WNT signaling for 12 hours (which generates posterior epiblast cells), the continued activation of WNT generated primitive streak cells. By contrast, WNT **inhibition** generated PGCLCs. This result was confirmed across three additional hESC lines (H1, H9, and H9-NANOG-YFP; new data available in Fig. S1).

It should also be noted that we tried two conditions: simply omitting exogenous WNT from the culture, or actively suppressing WNT signaling by small molecules. Active WNT suppression generated PGCLCs more efficiently than simply omitting WNT: this implies that some cells in the culture endogenously produce WNTs, thus reducing the efficiency of differentiation. However, given that we cannot achieve 100% efficient PGCLC induction, we conclude that WNT inhibition is necessary but not sufficient for PGCLC induction. Other signaling pathways, yet to be discovered, likely also have an important role.

2. The authors also found that NANOG and OCT4 was continuously expressed during the course of PGCLC differentiation from hESCs (Fig. 5F, Fig. S5, page 8-9), but functional significance of their expression is not shown.

Response: We thank the reviewer for this comment. We agree on the importance of understanding how and why NANOG and OCT4 are continuously expressed during PGCLC differentiation. However, this is beyond the scope of this manuscript and certainly warrants further investigation.

3. Previous studies showed that hESC-derived PGCLCs were efficiently purified by using combination of cell-surface markers such as EpCAM and Integrin $\alpha 6$, and merits of the presented methods in this study is unclear.

Response: We found that previously-published surface markers such as EPCAM/CD326 and ITGA6/CD49F were indeed expressed on PGCLCs; however they were also expressed on non-PGCLCs (e.g., hPSCs) and were

Fig. S4 (new): Single-cell RNA-seq of surface marker expression across cell-types generated in this study

therefore less specific (**Fig. S4**). The original study reporting that PGCLCs were EPCAM+ ITGA6+ likewise showed that ~26% of undifferentiated hPSCs were also EPCAM+ ITGA6+ (Fig. 5b in Sasaki et al., 2015; *Cell Stem Cell*).

The CXCR4+ PDGFRA- GARP- surface marker combination appears to identify PGCLCs more precisely. We rigorously confirmed that FACS-sorted CXCR4+ PDGFRA- GARP- cells were indeed highly enriched for PGCLCs using both immunostaining (**Fig. 4b**) and single-cell RNA-seq (**Fig. 5d**).

We hope that this CXCR4+ PDGFRA- GARP- surface marker combination will constitute a new tool in the hands of investigators interested in germ cell biology. It should be noted that CXCR4 is a well-recognized, evolutionarily-conserved surface marker of PGCs in zebrafish and mouse, and we extend this marker to human.

4. Although the authors mentioned in page 4 that ‘However, D0.5 cells expressed posterior epiblast/primitive streak markers at lower levels compared to D1 primitive streak cells that were generated by 24 hours exposure to posteriorizing signals (Fig. 1e)’, the expression of FGF8 is not lower in the D0.5 cells than that in D1 PS cells.

Response: Thank you for pointing this out. We have added “with the exception of FGF8” in the text.

5. In Fig. S2B, please show the effect of the conventional concentration of BMP4 (200 ng/ml) for PGCLC induction in addition to that of the lower concentration (20 ng/ml).

Response: In **Fig. S2b**, we tested 3 different BMP4 concentrations, up to 50 ng/mL. Since we are observing satisfactory results with lower concentrations, we did not include higher concentrations of BMP4 in our tests.

6. In page 5, the authors mentioned that ‘PGCLCs generated using our protocol did not express markers of endoderm (FOXA2, HHEX) or extraembryonic fate (CDX2), thus reaffirming their lineage specificity (Fig. S2e)’, but Fig.S2e seems to show their expression in D3.5 PGCLCs. They also said that in the same page that ‘Our monolayer differentiation protocol generated SOX17-GFP+ PGCLCs that expressed NANOS3 but not endodermal marker FOXA2 (Fig. S2f,g)’, but Fig.S2F again shows significant expression of FOXA2 in D3.5 PGCLCs.

Response: The qPCR plots may have given the inaccurate impression that endodermal markers were expressed in PGCLCs, because the y-axis was normalized to levels found in undifferentiated hPSCs (i.e., hPSC expression level = 1.0). Because hPSCs express negligible levels of these markers, small increases in these markers relative to hPSCs does not necessarily indicate significant upregulation of these markers. In our single-cell RNA-seq, we included a positive control (hPSC-derived definitive endoderm cells). This showed that PGCLCs minimally express endodermal markers by comparison to actual endodermal cells (**Fig. 5e, Fig. S5d**).

7. In page 6, the authors mentioned that ‘the mesodermal markers PDGFR α /CD140A (Kattman et al., 2011) and GARP/LRRC32 (Loh et al., 2016) were expressed on the D3.5 non-PGCLCs (Fig. 3a,b)’, but Fig.3b shows no PDGFR α -positive cells in mCherry-negative cells at least by the presented gating.

Response: We thank the reviewer for pointing this out. In certain hPSC lines, we find that a population of PDGFRA+ non-PGCLCs is erroneously produced (e.g., **Fig. 3c**). Therefore for maximum stringency we select for PDGFRA- GARP- cells in order to rigorously exclude non-PGCLCs.

8. In the last paragraph in page 6, the authors mentioned that ‘hPSC-derived CXCR4+ PDGFR α - GARP- PGCLCs upregulated of hallmark PGC markers without substantial expression of endodermal or mesodermal 6 markers (Fig. 4b, Fig. S3b)’, but the figures show substantial expression of the somatic markers in D3.5 PGCLCs compared with that in ESC and D3.5 bulk cells.

Response: The qPCR plots may have given the inaccurate impression that endodermal markers were expressed in PGCLCs, because the y-axis was normalized to levels found in undifferentiated hPSCs (i.e., hPSC expression level = 1.0). Because hPSCs express negligible levels of these markers, small increases in these markers relative to hPSCs does not necessarily indicate significant upregulation of these markers. Our single-cell RNA-seq analysis is more comprehensive, and shows minimal to no levels of endodermal or mesodermal marker expression (**Fig. 5e, Fig. S5d**).

9. In Fig.6, comparison of transcriptome of D3.5 PGCLCs in the monolayer culture with that of PGCLCs in 3D aggregates should be informative to discuss their possible differences.

Response: Thank you for this suggestion. We have now compared single-cell RNA-seq datasets, and find that PGCLCs generated in our simplified 2D protocol are transcriptionally similar to PGCLCs generated in a previous 3D protocol (by Amander Clark's group; Chen et al., 2019; *Cell Reports*) (**Fig. S10**). Our PGCLCs cluster together with Chen et al.'s PGCLCs (**Fig. S10a**), and show a high transcriptional correlation with them ($r = 0.91$, **Fig. S10g**).

10. The authors claimed that 'Single-cell RNA-seq analysis also revealed strong transcriptome wide similarities between hPSC-derived CXCR4+ PDGFR α -GARP- PGCLCs and early human fetal PGCs (Li et al., 2017)' in page 11, but the transcriptome in Fig. 6 shows distinct features of D3.5 PGCLCs and of FGCs.

Response: Thank you for raising this point. As the reviewer aptly pointed out, *in vitro* PGCLCs and *in vivo* FGC1 are not identical, as they likely reflect different developmental stages: the FGC1 analyzed by Li et al. have already colonized the fetal gonad, whereas *in vitro* PGCLCs likely correspond to pre-migratory PGCs that are incipiently emerging from pluripotent cells. PGCLCs do not yet express more "mature" germ cell markers such as *DAZL* (**Fig. 6e**). It is also notable that fetal germ cells such as FGC1 have been induced, migrated, and colonized the gonads in a "perfect" *in vivo* environment. We now explicitly address this in the main text.

We provide additional characterization of the genes that are differentially expressed between *in vitro* PGCLCs and *in vivo* FGC1 (and their associated gene ontology terms) in **Fig. S9**.

REVIEWER COMMENTS

Reviewer #1 (Remarks to the Author):

In this revision, authors added some more data to further support author's previous conclusion. However, my overall impression on this manuscript remains the same: it has a critical dearth of originality and novelty. Authors reached the same feat using 2D method as what has been already achieved by 3D method over last 6 years by a number of studies (Irie et al. 2015, Kobayashi et al. 2017, Sasaki et al. 2015, Chen et al. 2019, Kojima et al. 2017). What is the significance of 2D method over 3D method? Is there any analysis that can only be done by using 2D method? Authors could have provided such proof of concepts but throughout the manuscript, this point has not been addressed. 3D methods is highly efficient (30-60% induction), scalable, and amenable to various functional assays (Chen et al. 2019, Irie et al. 2015, Kobayashi et al. 2017, Sasaki et al. 2015, Yamashiro et al. 2018). Moreover, PGCLCs obtained by 3D method enabled further differentiation of these germ cells (Yamashiro et al. Science 2018), which has not been achieved by 2D method. Requirement of WNT signaling pathway has been extensively characterized (Kobayashi et al., 2017, Sasaki et al., 2015, Kojima et al., 2017). Subsequent inhibition of WNT signaling might be the point that author want to emphasize here but the mechanisms behind the role of WNT inhibition in the fate decision between PS versus PGCLC has not been addressed in this study. Moreover, PGCLCs can be obtained by 3D method, in high efficiency, without such WNT inhibition, raising the significance of WNT inhibition somewhat dubious.

There are some more critical points that authors failed to address in this revision:

1. Continuous NANOG expression during germ cell fate determination: As authors are aware of, this is conceptually not new. A previous paper in cynomolgus monkey PGCs suggest the continuous expression of NANOG (Sasaki et al. 2016). Moreover, previous 3D method validated the expression of pluripotency associated genes, including OCT4 and NANOG along time course (d0,2,4,6,8, Fig.2, Sasaki et al. 2015). These studies have not analyzed gene expression in single cell resolution, but it is obvious that these genes do not drop acutely between short intervals that has not been sampled. Moreover, it is unclear why continuous NANOG expression during the fate transition is so important. As reviewer#3 suggested, authors could have provided the functional significance (Reviewer #3 comment 2), which authors failed to provide in this revision.

2. Authors emphasize the significance of single cell analysis of PGCLCs. Again, single cell characterization of PGCLCs has shown the heterogeneity and gene expression dynamics during the fate specification (Chen et al. Cell Reports 2019). Characterization of non-PGC population that processes the mesodermal/endodermal fate has also been identified (Irie et al. 2015, Kojima et al. 2017). Comparison between PGCLCs and human germ cells in vivo has also been done by both bulk RNA-seq (Irie et al. 2015 Cell) and single cell RNA-seq (Hwang et al. 2020 Nat Commun).

3. Previous surface markers (ITGA6 and EPCAM) successfully isolate PGCLC from non-PGCLC with high accuracy (~99% of cells showing BLIMP1-tdTomato and TFAP2C-EGFP) (Sasaki et al 2015). Author pointed out that these markers are expressed in PSCs and therefore not-specific. However, the purpose of surface markers is to separate the particular subset of cells from unwanted cell types present in the sample of interest. When the PGCLC is formed, these surface markers could sufficiently eliminate these unwanted population with extremely high accuracy. So what is the point to compare these surface marker expression with those of PSCs? Teratoma can be formed if undifferentiated PSCs are contaminated in the sample, which can be a serious concern under a clinical setting. This possibility might merit further investigation but has not been addressed in this present study, either. In fact, only~90% of sorted cells by CXCR4+GARP-PDGFRα- cells express BLIMP1/SOX17 (Fig. 4B), raising the concern that the non-PGCs are still present in a significant fraction of sorted cells.

4. Line 157, "assignment of terms such as "posterior epiblast" or "primitive streak" in human in premised on evolutionary homology to other mammals such as pig and mouse epiblast cell,..." these are terms referring to anatomic structures on post-implantation embryos, which are shown to be highly divergent between primates and other species (e.g. mice, pig). Therefore, "evolutionary

homology" is not substantiated.

Reviewer #2 (Remarks to the Author):

This revised manuscript from Vittorio Sebastiano's group reports a novel 2D/monolayer approach to in vitro differentiation of human primordial germ cell like cells (PGCLCs) from pluripotent stem cells. I felt this paper was very strong initially and am even more in favor after extensive good-faith revisions that have addressed nearly all of my minor criticisms from the initial version. I especially appreciate the new data that show repeatability in different lines, improve depth of characterization, compare to existing methods, connect "developmental" time points, and add text that clarifies misunderstandings/misconceptions/unclear outcomes. The only point not addressed to satisfaction is the exclusive use of pseudotime for trajectory inference whereas I asked for RNA velocity analysis which is more robust and informative. This is a relatively minor point and should not preclude publication. I have no further criticisms.

Reviewer #3 (Remarks to the Author):

The revised manuscript has been improved in the data concerning transcriptional features of the PGCLCs induced by the monolayer method described in this study, including their similarity to those of PGCLCs induced by the conventional 3D culture method and of in vivo human PGCs. Meanwhile mechanistic insights of the reduced WNT signal and functional significance of continuous expression of the pluripotency genes in PGCLC specification were not addressed, which is disappointing.

Importance of reduced WNT signal after its transient activation during PGCLC specification was clearly demonstrated in this study, which is very nice. The authors discussed that continuous WNT signal induces primitive streak fate at the expense of PGC fate, but it is totally unclear why PGC fate is enhanced when primitive streak fate is suppressed. Examining changes of gene expression by single cell transcriptome and identify candidate downstream genes by WNT inhibition and by its activation in different time points after Day 0.5 should provide crucial cues for this issue.

Continuous expression of the pluripotency genes does not necessarily mean its functional importance during PGC specification. Its functional importance is assessable by temporal downregulation of the candidate pluripotency genes by an inducible knock-down vector.

Above mentioned additional experiments should provide novel mechanistic insight for human PGC specification, which substantially improves this manuscript.

Additional comments;

1. The author tested a relatively narrow range of low BMP4 concentration for human PGC specification, and it is unclear whether the conventional higher concentration of BMP4 results in more efficient PGC formation. Please address this point.
2. Because about 90% or more of CXCR4 positive cells were always GFAP and PDGFRa double negative, sorting by CXCR4 seems to be enough to obtain pure PGCs. Please explain more clearly a merit of using the additional negative markers and importance to obtain even slightly pure PGCs.

RESPONSE TO REVIEWERS' COMMENTS

We thank the reviewers for their valuable and constructive comments. We have conducted extensive additional experiments and bioinformatics analysis that confirm, validate, and further enhance our original observations. We hope the reviewers will find the revised manuscript acceptable for publication.

Below we provide the original reviewers' comments (in **blue**) and our responses (in black).

Reviewer #1 (Remarks to the Author): In this revision, authors added some more data to further support author's previous conclusion. However, my overall impression on this manuscript remains the same: it has a critical dearth of originality and novelty. Authors reached the same feat using 2D methods what has been already achieved by 3D method over last 6 years by several studies (Irie et al. 2015, Kobayashi et al. 2017, Sasaki et al. 2015, Chen et al. 2019, Kojima et al. 2017). What is the significance of 2D method over 3D method? Is there any analysis that can only be done by using 2D method? Authors could have provided such proof of concepts but throughout the manuscript, this point has not been addressed.

We thank the reviewer for the comments; we are puzzled about the fact that the novelty and significance of this work is not fully grasped and appreciated.

Novelty of the study is on many fronts:

- First monolayer differentiation platform capable of generating yields of hPGCLCs comparable and even more efficient than 3D culture systems.
- Demonstration that WNT signaling needs to be dynamically activated to promote PGCLCs differentiation from hPSCs: initial activation (optimally for 12 hrs) followed by sharp inhibition. It should be noted that since we can precisely measure the effects of one signaling pathway at a time, we can determine not only the necessity but also the sufficiency of it. As an example, despite this novel discovery on the role of dynamic modulation of WNT, we still cannot achieve 100% purity of differentiation. This suggests that other yet TBD signaling pathways are necessary. Our platform allows for systematic interrogation of other signaling pathways that other protocol cannot easily allow for.
- Demonstration that NANOG and OCT4 are specifically expressed in cells that progressively mature to PGCLCs state and demonstration that at least NANOG is required (new data generated for this current resubmission)
- Demonstration that a bipotent population of cells (that we name here posterior epiblast) can generate both PS and PGCs by in part modulation of WNT signaling. This solves a long-standing question in the field: are PGCs re-acquiring a "quasi-pluripotent" gene expression profile after having transitorily converted to a somatic state (where pluripotency markers are temporarily silenced) or is expression of pluripotency markers continued to be expressed in cells that progressively acquire a PGCs fate? We demonstrate that the second hypothesis is the correct one.

Ample literature has been generated discussing the advantages of 2D vs 3D culture conditions (and vice versa).

Among the advantages of a 2D culture systems are the following:

- Cells in monolayer (2D) are exposed to homogenous culture conditions and homogeneous concentration of small molecules, nutrients and signaling proteins. Conversely, 3D cultures are exposed to gradients of the above and cells exposed to varying concentrations that can elicit a very diverse response in different cell types which cannot be controlled in a standardized and rigorous manner. In addition, 2D cultures can be rapidly screened for various combinations of growth conditions. In our study we have taken advantage of this and shown that WNT signaling is highly dynamic. After an initial activation, WNT inhibition is necessary for PGC formation. We have shown that BMP concentration can be significantly lowered in 2D cultures with an impact on reproducibility and costs. We have shown that BMP is dispensable on day 1.5-2.5.
- Cells in monolayer can be rapidly genetically tested with siRNAs for a variety of different genes without recurring to genetic manipulation of cells that are costly, time consuming, and can lead to unwanted and uncontrolled genetic alterations of the cells that can impact their behavior. This represents a huge advantage because it allows rapid screening of genetic contribution to a specific biological question/phenomenon. In this study we have shown that NANOG expression is important for generation of hPGCs. New data generated for this current resubmission shows that NANOG knock-down as early as day 0 of differentiation has a profound effect on the percentage of PGCLCs generated at day 3.5, with as much as ~5-fold reduction in PGCLCs generated.
- 2D methods are amenable to micropatterned culturing methods, which are ideal for the study of differentiation spatial dynamics and colony size in improving the efficiency of *in vitro* generation of specific cell types (shown by various studies, including Jo et al, *eLife* **11**:e72811 (2022) for PGCLCs, Warmflash A et al Nature Methods **volume 11**: 847–854 (2014) -to recapitulate early embryonic patterning using micropatterned hESC differentiation cultures).

3D methods are highly efficient (30-60%induction), scalable, and amenable to various functional assays (Chen et al. 2019, Irie et al. 2015, Kobayashi et al. 2017, Sasaki et al. 2015, Yamashiro et al. 2018).

In our hands our monolayer differentiation platform is far superior to 3D methods. Please see below.

Moreover, PGCLCs obtained by 3D method enabled further differentiation of these germ cells (Yamashiro et al. *Science* 2018), which has not been achieved by 2D method.

Further follow up studies will address this. Our data (generated with several distinct and complementary approaches) demonstrate that the cells that we generate with our platform are *bona fide* early PGCLCs that are transcriptionally similar to mitotically active human fetal germ cells. We have also benchmarked our PGCLCs to PGCLCS obtained by other methods (3D cultures) and confirmed the germ cells nature of our cells. We are conducting follow up studies on further maturation of PGCLCs, but this is beyond the scope of the current study. The deep, rigorous, and thorough molecular and

transcriptional characterization we have performed indicate that we can generate *bona fide* human PGCLCs.

Requirement of WNT signaling pathway has been extensively characterized (Kobayashi et al., 2017, Sasaki et al., 2015, Kojima et al., 2017). Subsequent inhibition of WNT signaling might be the point that author want to emphasize here but the mechanisms behind the role of WNT inhibition in the fate decision between PS versus PGCLC has not been addressed in this study.

The importance of WNT has been previously demonstrated. But the dynamic requirement of this has not been addressed. Here we show that for successful PGCLCs formation initial WNT activation (ideally for 12 hrs) must be followed by inhibition of WNT.

Further data generated for this resubmission show some mechanistic insights on the role of WNT signaling. We demonstrate very clearly that after the initial activation of WNT, if WNT signaling continues to be activated it leads to formation of Primitive streak cells. If, conversely, the WNT activation is actively inhibited it leads to a more efficient formation of PGCLCs. This finding is new and points to a new model where a bipotent population of cells (we refer to them as posterior epiblast) can give rise to both PS and PGCLCs and this in part is explained by mutually exclusive fates in part dictated by WNT signaling modulation. Please see below for detailed description on our new findings supporting this.

Moreover, PGCLCs can be obtained by 3D method, in high efficiency, without such WNT inhibition, raising the significance of WNT inhibition somewhat dubious.

PGCs have been obtained efficiently with several different approaches. This does not undermine the value of having alternative approaches and solutions that are equally (or more) efficient, scalable, highly reproducible, and more standardized.

The importance of WNT inhibition is new and is not dubious but rather supported by a wealth of data and complementary approaches in our study. The role of WNT inhibition was not previously identified simply because it was not studied. Is the reviewer implying that any new discovery should be considered dubious simply because of its intrinsic novel aspect?

We have now performed additional experiments with and without WNT manipulation (see below) to demonstrate the significance of WNT pathway inhibition in generating PGCLCs with better efficiency. Please see below for detailed description on our new findings supporting this.

There are some more critical points that authors failed to address in this revision:1. Continuous NANOG expression during germ cell fate determination: As authors are aware of, this is conceptually not new. A previous paper in cynomolgus monkey PGCs suggest the continuous expression of NANOG (Sasaki et al. 2016). Moreover, previous 3D method validated the expression of pluripotency associated genes, including OCT4 and NANOG along time course (d0,2,4,6,8, Fig.2, Sasaki et al. 2015). These studies have not analyzed gene expression in single cell resolution, but it is obvious that these genes do not drop acutely between short intervals that has not been sampled. Moreover, it is

unclear why continuous NANOG expression during the fate transition is so important. As reviewer#3 suggested, authors could have provided the functional significance (Reviewer#3 comment 2), which authors failed to provide in this revision. 2. Authors emphasize the significance of single cell analysis of PGCLCs. Again, single cell characterization of PGCLCs has shown the heterogeneity and gene expression dynamics during the fate specification (Chen et al. Cell Reports 2019). Characterization of non-PGC population that processes the mesodermal/endodermal fate has also been identified (Irie et al. 2015, Kojima et al. 2017). Comparison between PGCLCs and human germ cells in vivo has also been done by both bulk RNA-seq (Irie et al. 2015 Cell) and single cell RNA-seq (Hwang et al. 2020 Nat Communications).

The reviewer is certainly aware that cynomolgous monkeys are not human and that hominoids and old-world monkeys diverged in evolution approximately 25 million years ago. If we were to simply rely on animal models and assume that development is identical in different animal species, there would be no need to model human-specific aspects of development and genetics using human stem cells. As an example, in cynomolgous monkeys, data suggest an amniotic origin of PGCs. Does this mean that the same is true for human development? OCT4 and NANOG expression has been shown in bulk analysis in previous works. We show here that NANOG and OCT4 are specifically co-expressed in cells that differentiate into PGCLCs, and information at a single cell resolution is nonetheless important to be shown and not to be assumed based on bulk data. In addition, we show that NANOG is necessary for PGCs formation. In this revision, we provide evidence for a functional requirement for NANOG expression in PGCLC generation. Knock-down of NANOG using siRNA, as early as Day0 or Day0.5 of differentiation led to a 5-fold reduction in PGCLCs generated. Knock-down of NANOG on all subsequent days of Day1.5 and Day2.5 similarly showed reduction in PGCLC efficiency, albeit a milder one. And knock-down on Day3.5, and cells harvested a day later, still showed between ~2-5-fold reduction in PGCLC generation. From these results, we conclude that there is a clear early and persistent requirement for NANOG in PGCLC specification, and one which supports our observation of continuous NANOG expression in the transition from pluripotent cells to PGCLC identity (Fig 8E and S11C, pasted below for your convenience).

E Early and persistent NANOG expression is essential for PGCLCs generation

C NANOG expression in NANOG knock-down cells (related to Fig. 8D)

3. Previous surface markers (ITGA6 and EPCAM) successfully isolate PGCLC from non-PGCLC with high accuracy (~99% of cells showing BLIMP1-tdTomato and TFAP2C-EGFP) (Sasaki et al 2015). Author pointed out that these markers are expressed in PSCs and therefore not-specific. However, the purpose of surface markers is to separate the particular subset of cells from unwanted cell types present in the sample of interest. When the PGCLC is formed, these surface markers could sufficiently eliminate these unwanted population with extremely high accuracy. So what is the point to compare these surface marker expression with those of PSCs? Teratoma can be formed if undifferentiated PSCs are contaminated in the sample, which can be a serious concern under a clinical setting. This possibility might merit further investigation but has not been addressed in this present study, either. In fact, only ~90% of sorted cells by CXCR4+GARP-PDGFR α - cells express BLIMP1/SOX17 (Fig. 4B), raising the concern that the non-PGCs are still present in a significant fraction of sorted cells.

The point we are making is that we have found three novel markers that can be used for PGCLCs differentiation. Having these markers is an additional set of tools that can benefit the scientific community. The reviewer should also consider the value of having found that, like in many other species, CXCR4 seems to be a conserved marker of PGCs formation. It should also be noted that our protocol is more defined not only in generating PGCLCs but also in generating contaminating cells. Hence the value of having additional markers that help eliminate contaminating cells that may be different from contaminating cells obtained with less defined 3D protocols.

It should be noted that the cells are stained post-sorting. Some of the cells get mechanically damaged and so, despite showing nuclear staining may have a low or neglectable level of expression of PGCs markers.

Our Bulk RNA-seq analysis also clearly shows that the combination of CXCR4+GARP-PDGFR α - is superior to ITGA6+EPCAM+ combination, whereby the latter is insufficient to confidently distinguish hPSCs from PGCLCs (Fig. S10D).

D Surface marker combination for sorting PGCLCs

4. Line 157, “assignment of terms such as “posterior epiblast” or “primitive streak” in human in premised on evolutionary homology to other mammals such as pig and mouse epiblast cell,...” these are terms referring to anatomic structures on post-implantation embryos, which are shown to be highly divergent between primates and other species (e.g. mice, pig). Therefore, “evolutionary homology” is not substantiated.

This statement conflicts with the comment made above on cynomolgus monkeys. We conclude that the reviewer values the identification of species-specific characters of development. We are using the term posterior epiblast because regardless of

conservation, there is always an anterior and posterior side of the embryo. The initial phase of differentiation is used as a posteriorizing signal, hence the use of the term “posterior epiblast”. Anteriorizing signals are different. We welcome suggestions on the use of alternative terminology the reviewer feels more comfortable with.

Reviewer #2 (Remarks to the Author): This revised manuscript from Vittorio Sebastiano’s group reports a novel 2D/monolayer approach to in vitro differentiation of human primordial germcell like cells (PGCLCs) from pluripotent stem cells. I felt this paper was very strong initially and am even more in favor after extensive good-faith revisions that have addressed nearly all of my minor criticisms from the initial version. I especially appreciate the new data that show repeatability indifferent lines, improve depth of characterization, compare to existing methods, connect “developmental” time points, and add text that clarifies misunderstandings/misconceptions/unclear outcomes. The only point not addressed to satisfaction is the exclusive use of pseudotime for trajectory inference whereas I asked for RNA velocity analysis which is more robust and informative. This is a relatively minor point and should not preclude publication. I have no further criticisms.

We thank the reviewer for the enthusiasm and for appreciating the value and the novelty of our work.

Reviewer #3 (Remarks to the Author): The revised manuscript has been improved in the data concerning transcriptional features of the PGCLCs induced by the monolayer method described in this study, including their similarity to those of PGCLCs induced by the conventional 3D culture method and of in vivo human PGCs.

We thank the reviewer for the overall positive feedback about our study.

Meanwhile mechanistic insights of the reduced WNT signal and functional significance of continuous expression of the pluripotency genes in PGCLC specification were not addressed, which is disappointing.

We agree, and we have performed extensive additional experimental work to address this important aspect (please see below)

Importance of reduced WNT signal after its transient activation during PGCLC specification was clearly demonstrated in this study, which is very nice.

Thank you.

The authors discussed that continuous WNT signal induces primitive streak fate at the expense of PGC fate, but it is totally unclear why PGC fate is enhanced when primitive streak fate is suppressed. Examining changes of gene expression by single cell transcriptome and identify candidate downstream genes by WNT inhibition and by its activation in different timepoints after Day 0.5 should provide crucial cues for this issue.

We have addressed this important point by performing bulk RNASeq at every step of the differentiation and included sorted PGCs (please see below). We gained important clues on the mechanistic role of WNT and we hope the reviewer is now satisfied with our characterization.

Continuous expression of the pluripotency genes does not necessarily mean its functional importance during PGC specification. Its functional importance is assessable by temporal downregulation of the candidate pluripotency genes by an inducible knock-down vector.

We performed additional experiments (see below) to address this important point.

Above mentioned additional experiments should provide novel mechanistic insight for human PGC specification, which substantially improves this manuscript.

In this resubmission, we have tackled the important and still outstanding points raised by the reviewer and we are hopeful we have now fully addressed them.

The authors discussed that continuous WNT signal induces primitive streak fate at the expense of PGC fate, but it is totally unclear why PGC fate is enhanced when primitive streak fate is suppressed. Examining changes of gene expression by single cell transcriptome and identify candidate downstream genes by WNT inhibition and by its activation in different timepoints after Day 0.5 should provide crucial cues for this issue.

Our model is that at each point of cell fate commitment, there is a bifurcation of cell fate. We hypothesized that in addition to providing activating signals for cell fate of interest, to have a better efficiency, suppression of the opposing cell fate choice is required. We have now performed additional experiments as requested by the reviewer to address the importance of continued WNT suppression (as opposed to no WNT activation or no WNT manipulation) in PGCLC generation.

We have now performed bulk RNA-seq analyses from different time-points of PGCLC differentiation with and without WNT pathway manipulation (Day1, Day1.5, Day2.5, Day3.5 and Day3.5 NANOG+ sorted PGCLCs and NANOG- populations) – **Fig.8 and S11.**

Using this approach, we have now provided a comprehensive and additional wealth of data that explains:

- 1) That WNT inhibition (“**XAV939**”) is required after D0.5, from Day1 to Day1.5, to suppress a primitive streak fate (**Fig.S11**, pasted below for your convenience). We show that i) continued WNT activation (“**CHIR99021**”) promotes expression of pan-primitive streak markers (*T*, *MIXL1*) and posterior primitive streak (*EVX1*, *MESP1*), and to a lesser extent anterior primitive streak (*EOMES*, *GSC*); ii) lack of any WNT manipulation (“**Base media**”) results in reduced primitive streak markers and iii) this reduction is further enhanced with WNT pathway inhibition (XAV939).

Fig. S11

- 2) WNT inhibition further promotes activation of PGC marker genes (e.g., *NANOG*, *POU5F1*, *TFAP2C*, *SOX17*, *PRDM1*(*BLIMP1*)), and to higher levels than with no WNT manipulation. Moreover, XAV939 treatment further suppressed mesodermal markers (*ACTC1*, *TMEM88*) better than without any WNT manipulation (**Fig.8C**, pasted below for your convenience)).

- 3) WNT pathway suppression promotes significantly better efficiency of PGCLC generation at Day 3.5 (~30% with Base media vs. 50% with XAV939) (**Fig.8B**, pasted below for your convenience).

4) Pearson correlation analysis revealed that NANOG+ sorted PGCLCs from Day3.5 XAV939 and D3.5 Base media are highly correlated ($r=0.99$). However, sorted PGCLCs are better correlated with unsorted cells from Day3.5 XAV939 ($r=0.65$) but poorly correlated with unsorted cells from D3.5 Base media ($r=-0.05$), further confirming that XAV939 promotes better efficiency PGCLC generation (**Fig.8D**, pasted here for your convenience and **S10C**).

D Pearson correlation of Day3.5 sorted NANOG+ cells with and without WNT pathway manipulation

5) By comparing NANOG+ populations from Day3.5 (XAV939) vs. Day3.5 (Base media), we also show that the former is

required to maintain lower WNT ligand expression (*WNT5B*) and consequentially lower WNT pathway activity in PGCLCs (WNT target genes: *LEF1*, *SP5*) (**Fig.9**, pasted here for your convenience), thus contributing to a higher efficiency of PCGLC generation with WNT inhibition (**Fig. 8B & D**). Differential gene expression analysis

F Differentially expressed genes in sorted NANOG+ cells from Day3.5 XAV939 vs. Base media treatment

showed *WNT5B* to be significantly upregulated in Base media condition (**Fig.S10F**).

- 6) Comparison of NANOG⁺ vs. NANOG⁻ populations from XAV vs. Base media shows that *WNT5B* is expressed at a higher level in NANOG⁻ fraction, which also show higher WNT activity (*LEF1* and *SP5*). However, with XAV treatment, *WNT5B* is reduced in NANOG⁺ PGCLCs, and correspondingly reduced WNT activity in NANOG⁺ and NANOG⁻ cells (*LEF1* and *SP5*). From this we infer that continued WNT suppression has a role in reducing unwanted WNT ligand expression in the culture that can promote unwanted WNT pathway activation in NANOG⁺ and NANOG⁻ cells that can inhibit PGCLC generation (**Fig.9**, pasted below for your convenience).

Continuous expression of the pluripotency genes does not necessarily mean its functional importance during PGC specification. Its functional importance is assessable by temporal downregulation of the candidate pluripotency genes by an inducible knock-down vector. Above mentioned additional experiments should provide novel mechanistic insight for human PGC specification, which substantially improves this manuscript.

We have now performed NANOG knock-down (with siRNA) experiment and show that NANOG is required from an early time-point and also continuously till Day 3.5 to ensure efficient generation of PGCLCs. We used CXCR4+NANOG-YFP⁺ expression to measure PGCLC levels and show that early knock-down of NANOG reduced the efficiency from ~50% to <20% (**Fig.8E**, pasted below for your convenience).

E Early and persistent NANOG expression is essential for PGCLCs generation

NANOG expression in NANOG KD assay

Additional comments;1. The author tested a relatively narrow range of low BMP4 concentration for human PGC specification, and it is unclear whether the conventional higher concentration of BMP4 results in more efficient PGC formation.

We have conducted the suggested experiment using 20, 40, 50, 100, and 200 ng/mL of BMP. As it is clear (chart pasted below), standard concentration (200 ng/mL) of BMP4 does not augment the percentage of PGCLs. For simplicity we have kept the original figure in the manuscript.

Please address this point.2. Because about 90% or more of CXCR4 positive cells were always GFAP and PDGFR α double negative, sorting by CXCR4 seems to be enough.

As we have shown in **Fig. S10D** (pasted below for your convenience), contaminating mesoderm-like (NANOG-) populations express PDGFR α and segregate better with PDGFR α and such dual-marker selection strategy might be better for a more cleaner sorting of PGCLCs.

D Surface marker combination for sorting PGCLCs

REVIEWERS' COMMENTS

Reviewer #1 (Remarks to the Author):

In this revision, authors added some data to support author's previous conclusion. The reviewer understand that authors want to emphasize the differences of this study from previous studies, but these are incremental at best and do not provide major conceptual advances to the field. As the reviewer commented previously, the role of Wnt signaling in hPGCLC induction process has been well described. The author's findings related to the "Dynamic requirement of Wnt" seems incremental and do not add much values to the previously known "role of Wnt" on PGCLC induction. The reviewer understands that the role of Wnt inhibition was supported by data in this study and not previously described but the contribution of such finding to the field is probably small. The data that the extension of Wnt stimulation results in mesodermal differentiation has been previously shown (Kobayashi et al. 2017).

Besides, 2D induction of human PGCLCs have been published (Jo et al. eLife 2022;11:e72811.), showing the induction efficiency of ~70% in contrasts to ~20-30% in this paper. This paper needs to be cited properly. Overall, the reviewer acknowledge that there are a few new findings in this study here and there, but these findings are not well connected ("2D method", "role of dynamic requirement of Wnt", the role of NANOG) and the novelty is not sufficiently high.

Some of the "novelty" claims authors provided in RESPONSE TO REVIEWERS's COMMENTS do not appear to be well supported by evidence. Here are examples:

- First monolayer differentiation platform capable of generating yields of hPGCLCs comparable and even more efficient than 3D culture systems.

>The reviewer do not find strong evidence that 2D culture is more efficient than 3D culture. Fig.S1A might be the one but induction efficiency (4-12%) contradicts with those described in the original papers, which are expected to have 20-40% induction efficiency and have been reproduced in multiple independent laboratories. Moreover, the total number of PGCLCs have not been compared. It is possible that authors simply failed to reproduce the previous methods due to technical errors. Authors need to explain the discrepancy.

The reviewer appreciate that authors do live imaging for NANOG expression. However, as described by the previous comments, continued NANOG expression has already been suggested in human PGCLCs so it is not a surprising finding (authors did not respond to this comment made by the reviewer).

Fig. S10D: Expression levels among the replicates substantially varies, raising the question about the reproducibility of the induction method or reliability of the quantification. Please show what X and Y axis represent.

Page6, line 287: In contrast to this paper, human PGCLCs generated in previous 2D and 3D platform express T (Brachyury) (Jo et al. eLife 2022;11:e72811; Irie et al. Cell 2015; 160(1-2): 253, Sasaki etl al. 2015 Cell Stem Cell 7(2):178). Please explain the discrepancy.

Page8, line363-373/Fig. 8E, the differences do not appear to be statistically significant.

Page 8, line 372-373: The siRNA knockdown does not seem to be efficient. What is the transfection efficiency in 2D format versus 3D? Authors emphasize that transfection of siRNA is more challenging to perform in 3D but did not provide such evidence.

Page 26, line 1199-1211: please provide the detailed method of how the transfection was conducted.

Page10, line 475: Here is another study suggested otherwise so please integrate it in the discussion. Kojima et al. Cell Stem Cell. 2017 Oct 5;21(4):517-532.e5. doi: 10.1016/j.stem.2017.09.005.

"It should be noted that the cells are stained post-sorting. Some of the cells get mechanically damaged and so, despite showing nuclear staining may have a low or neglectable level of expression of PGCs markers."

>This is speculative and need to be supported by evidence.

This statement conflicts with the comment made above on cynomolgus monkeys. We conclude that the reviewer values the identification of species-specific characters of development.

>The reviewer mentioned that "post-implantation embryos, which are shown to be highly divergent between primates and other species (e.g. mice, pig)". Among primates (humans, cynomolgus monkeys), these structures are well conserved. Of course, that does not mean cynomolgus and humans are identical in germ cell development, but the reviewers' comments have not been conflicted each other. Authors should read the reviewer's comments more carefully.

Reviewer #3 (Remarks to the Author):

All of my comments for the previous version of the manuscript have been addressed in the current version, and the manuscript is substantially improved. An additional minor concern is the order of the figures, because Fig.8 and Fig.9 are mentioned after Fig.1 in the result section.

RESPONSE TO REVIEWERS' COMMENTS

We thank the reviewers for their comments. Below we provide the reviewers' comments (in blue) and our responses (in black).

Reviewer #1 (Remarks to the Author):

In this revision, authors added some data to support author's previous conclusion. The reviewer understand that authors want to emphasize the differences of this study from previous studies, but these are incremental at best and do not provide major conceptual advances to the field. As the reviewer commented previously, the role of Wnt signaling in hPGCLC induction process has been well described. The author's findings related to the "Dynamic requirement of Wnt" seems incremental and do not add much values to the previously known "role of Wnt" on PGCLC induction. The reviewer understands that the role of Wnt inhibition was supported by data in this study and not previously described but the contribution of such finding to the field is probably small. The data that the extension of Wnt stimulation results in mesodermal differentiation has been previously shown (Kobayashi et al. 2017). Besides, 2D induction of human PGCLCs have been published (Jo et al. *eLife* 2022;11:e72811.), showing the induction efficiency of ~70% in contrasts to ~20-30% in this paper. This paper needs to be cited properly. Overall, the reviewer acknowledge that there are a few new findings in this study here and there, but these findings are not well connected ("2D method", "role of dynamic requirement of Wnt", the role of NANOG) and the novelty is not sufficiently high.

We thank the reviewer for acknowledging that the explicit inhibition of WNT inhibition has "not [been] previously described". We also appreciate the reviewer for bringing up the Jo et al., 2022; *eLife* paper (PMID: 35394424), which we have now cited in our revised manuscript as a recent demonstration of human PGCLC induction in monolayer conditions.

Some of the "novelty" claims authors provided in RESPONSE TO REVIEWERS's COMMENTS do not appear to be well supported by evidence. Here are examples: • First monolayer differentiation platform capable of generating yields of hPGCLCs comparable and even more efficient than 3D culture systems.

>The reviewer do not find strong evidence that 2D culture is more efficient than 3D culture. Fig.S1A might be the one but induction efficiency (4-12%) contradicts with those described in the original papers, which are expected to have 20-40% induction efficiency and have been reproduced in multiple independent laboratories. Moreover, the total number of PGCLCs have not been compared. It is possible that authors simply failed to reproduce the previous methods due to technical errors. Authors need to explain the discrepancy.

Other laboratories have also shown that the efficiency of PGCLC specification in 3D culture can be lower than what the reviewer has suggested. For instance, another study showed that, on average, ~10% PGCLCs were produced across 11 different hPSC lines using the standard 3D differentiation method (Chang et al., 2021; *Cells*; PMID: 34572048).

The reviewer appreciate that authors do live imaging for NANOG expression. However, as described by the previous comments, continued NANOG expression has already been suggested in human PGCLCs so it is not a surprising finding (authors did not respond to this comment made by the reviewer).

We thank the reviewer for appreciating the live imaging results, which definitively show that NANOG is continuously expressed during PGCLC differentiation. As the reviewer pointed out, this was previously proposed, but not definitively shown.

Fig. S10D: Expression levels among the replicates substantially varies, raising the question about the reproducibility of the induction method or reliability of the quantification. Please show what X and Y axis represent.

As the reviewer has suggested, we have labeled Fig. S2D (formerly numbered as Fig. S10D) to indicate that it depicts mRNA expression of PGCLC surface-marker proteins.

Page6, line 287: In contrast to this paper, human PGCLCs generated in previous 2D and 3D platform express T (Brachyury) (Jo et al. eLife 2022;11:e72811; Irie et al. Cell 2015; 160(1-2): 253, Sasaki et al. 2015 Cell Stem Cell 7(2):178). Please explain the discrepancy.

We thank the reviewer for bringing up this very interesting point! *BRACHYURY/T* is a direct target gene of WNT signaling (Arnold et al., 2000; *Mech Dev*; PMID: 10704849). Therefore, our inhibition of WNT signaling may suppress *BRACHYURY* expression in PGCLCs produced in our protocol.

To address the reviewer's comment, the following has now been added to the main text: "*Of note, D3.5 PGCLCs generated in our system did not express BRACHYURY (Fig. 7e), which is expressed by PGCLCs generated by other differentiation systems (Irie et al., 2015; Jo et al., 2022; Sasaki et al., 2015). This may be explained by our inhibition of WNT signaling, as WNT is known to directly upregulate BRACHYURY expression (Arnold et al., 2000).*"

Page8, line363-373/Fig. 8E, the differences do not appear to be statistically significant.

As reported in Fig. 2e (formerly numbered as Fig. 8e), *NANOG* knockdown on D0 or D0.5 leads to a statistically significant decrease in PGCLC formation ($P=0.0003$ and $P=0.0027$, respectively). We have now included the tests for statistical significance in Fig. 2e.

Page 8, line 372-373: The siRNA knockdown does not seem to be efficient. What is the transfection efficiency in 2D format versus 3D? Authors emphasize that transfection of siRNA is more challenging to perform in 3D but did not provide such evidence.

We quantified the degree of *NANOG* knockdown by qPCR in our 2D culture system (Fig. S9C).

Page 26, line 1199-1211: please provide the detailed method of how the transfection was conducted.

Details for the siRNA transfection procedure are described in the "siRNA knockdown" section of the Methods, as follows:

"*NANOG-2A-YFP* hPSCs were transfected using ON-TARGETplus siRNA (Dharmacon) and RNAiMAX (Invitrogen) in 6-well plates, according to the manufacturer's instructions. In brief, 3 μ L RNAiMAX transfection reagent and 10 pmol siRNA were separately diluted in 50 μ L of Opti-MEM, and subsequently mixed and incubated for 5 minutes before dropwise addition to cells in 2 mL of the desired medium. Transfections were performed correspondingly with each media change in the described differentiation protocol: at day 0, 0.5, 1.5, 2.5, and 3.5. Samples were collected at each timepoint to assess the efficiency of knockdown by qPCR (as described above). For all siRNA transfection timepoints, the purity of PGCLCs was measured by flow cytometry at day 3.5 (the differentiation endpoint). Flow cytometry measurements were carried out by both (1) measuring YFP expression as a proxy for *NANOG* expression and (2) staining for CXCR4, which is specific to PGCLCs. $N=3$ biological replicates were used per group. Mean \pm SEM was calculated in GraphPad

Prism7 and statistical significance was calculated using Two-Way ANOVA with Šídák multiple test correction.”

Page10, line 475: Here is another study suggested otherwise so please integrate it in the discussion. Kojima et al. Cell Stem Cell. 2017 Oct 5;21(4):517-532.e5. doi: 10.1016/j.stem.2017.09.005.

This study has been cited in our manuscript at multiple junctures.

“It should be noted that the cells are stained post-sorting. Some of the cells get mechanically damaged and so, despite showing nuclear staining may have a low or neglectable level of expression of PGCs markers.”

>This is speculative and need to be supported by evidence.

The reviewer is speaking about our previous response to their comments (not the manuscript itself), so no changes to the manuscript have been made regarding this specific point. It should be noted that scRNA analysis confirmed 97% purity of sorted PGCLs.

This statement conflicts with the comment made above on cynomolgus monkeys. We conclude that the reviewer values the identification of species-specific characters of development.

>The reviewer mentioned that “post-implantation embryos, which are shown to be highly divergent between primates and other species (e.g. mice, pig)”. Among primates (humans, cynomolgus monkeys), these structures are well conserved. Of course, that does not mean cynomolgus and humans are identical in germ cell development, but the reviewers’ comments have not been conflicted each other. Authors should read the reviewer’s comments more carefully.

The reviewer is speaking about our previous response to their comments (not the manuscript itself), so no changes to the manuscript have been made regarding this specific point.

Reviewer #3 (Remarks to the Author):

All of my comments for the previous version of the manuscript have been addressed in the current version, and the manuscript is substantially improved. An additional minor concern is the order of the figures, because Fig.8 and Fig.9 are mentioned after Fig.1 in the result section.

Thank you! We have corrected the order of figures.